# FORGET VECTORS AT PLAY: UNIVERSAL INPUT PERTURBATIONS DRIVING MACHINE UNLEARNING IN IMAGE CLASSIFICATION

## ABSTRACT

Machine unlearning (MU), which seeks to erase the influence of specific unwanted data from already-trained models, is becoming increasingly vital in model editing, particularly to comply with evolving data regulations like the "right to be forgotten". Conventional approaches are predominantly model-based, typically requiring re-training or fine-tuning the model's weights to meet unlearning requirements. In this work, we approach the MU problem from an input perturbation-based perspective, where the model weights remain intact throughout the unlearning process. We demonstrate the existence of a proactive input-based unlearning strategy, referred to *forget vector*, which can be generated as an input-agnostic data perturbation and remains as effective as model-based approximate unlearning approaches. We also explore *forget vector arithmetic*, whereby multiple class-specific forget vectors can be combined through simple operations (*e.g.*, linear combinations) to generate new forget vectors for unseen unlearning tasks, such as forgetting arbitrary subsets across classes. Extensive experiments validate the effectiveness and adaptability of the forget vector, showcasing its competitive performance relative to state-of-the-art model-based methods while achieving superior parameter efficiency.

## 1 INTRODUCTION

To prevent unauthorized use of personal or sensitive data after training and comply with legislation such as the "right to be forgotten" (Hoofnagle et al., 2019), machine unlearning (**MU**) has garnered increasing attention as a solution to various challenges in vision tasks (Golatkar et al., 2020; Warnecke et al., 2021; Fan et al., 2023; Poppi et al., 2023). In essence, it initiates a reverse learning process to erase the impact of unwanted data (*e.g.*, specific data points, classes, or knowledge) from an already-trained model, while still preserving its utility for information not targeted by an unlearning request. Based on the guarantees provided for data removal from already-trained models, existing MU methods can be broadly categorized into two approaches: *exact unlearning* (Guo et al., 2019; Thudi et al., 2022b; Dong et al., 2024) and *approximate unlearning* (Izzo et al., 2021; Graves et al., 2021; Thudi et al., 2022a; Becker & Liebig, 2022; Chen et al., 2023b; Tarun et al., 2023; Cha et al., 2024). The former guarantees the complete and verifiable removal of targeted data, typically achieved by retraining the model from scratch with the data to be forgotten excluded from the training set, a process we refer to as *Retrain*. However, due to the high computational overhead, research has increasingly focused on approximate unlearning methods, which seek to achieve efficient unlearning without requiring full retraining.

Approximate unlearning strikes a balance between computational efficiency and effective data removal, making it practical for many real-world applications. Most existing approximate unlearning techniques are *model-based*, updating the model's weights within a limited number of training iterations to eliminate the influence of specific unwanted data, thus avoiding a full retraining process. Representative methods in this category include fine-tuning approaches (Warnecke et al., 2021; Perifanis et al., 2024), gradient ascent techniques (Thudi et al., 2022a; Chen et al., 2024), and influence function-based methods (Golatkar et al., 2020; 2021).

Although the model-based unlearning methods have made significant strides, they often overlook the *data-based* dimension and its potential impact on MU. For instance, it remains unclear whether current MU approaches generalize effectively to "shifted" forget data. Additionally, the possibility

of a data-based MU design that operates without updating model parameters has yet to be explored. Therefore, we ask:

> **(Q)** *Can we explore data influence in MU and harness data-based operations to fulfill MU?*

To address **(Q)**, we study MU from a fresh data-based viewpoint: **forget vector**, a universal input data perturbation designed to promote unlearning effectively; See the schematic overview in **Fig. 1**. Before developing the forget vector, we explore the rationale for how data perturbations complement current model-based MU approaches, as evidenced by these methods' generalization to common data shifts, including Gaussian noise and adversarial perturbations (Goodfellow et al., 2014; Hendrycks & Dietterich, 2019). To design the forget vector, we draw inspiration from recent input prompting techniques for vision models, known as visual prompting (Bahng et al., 2022b; Chen et al., 2023a; Oh et al., 2023) or model reprogramming (Elsayed et al., 2018; Zhang et al., 2022; Chen, 2024),

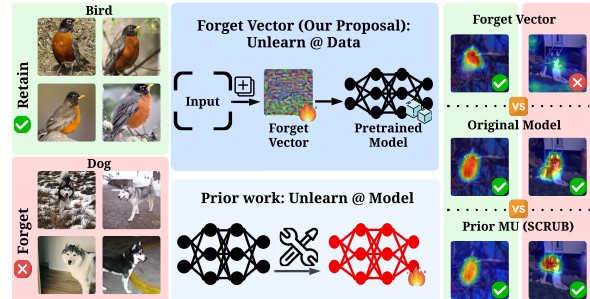

Figure 1: A schematic illustration comparing our proposed data-based MU method (termed the "forget vector"), which achieves unlearning objectives (*i.e.*, forgetting "dog" and remembering "bird" in this example) by operating directly on input data without altering model parameters, against traditional model update-based unlearning methods. ❌ indicates that the forget data is successfully unlearned, while ✅ means that the retain data is correctly recognized, or the forget data is *not* successfully unlearned. The "original model" refers to the model without unlearning applied, and "SCRUB" (Kurmanji et al., 2024b) is an existing representative unlearning method that updates model weights.

used in transfer learning and model adaptation. These prompting methods learn input perturbations to enable a fixed model to perform well on new tasks, effectively guiding the model to execute tasks it wasn't originally trained for. From this perspective, our research on the forget vector also explores whether it is possible to append a trainable "prompt" to the input to guide an already-trained neural network in unlearning specific data. The proposed forget vector allows the unlearner to modify user inputs targeted for deletion, offering a flexible and efficient approach to unlearning while potentially achieving significant parameter efficiency. We summarize **our contributions** below.

• We investigate the impact of forget data shifts on image classifiers post-unlearning, revealing that unlearning demonstrates resilience against these shifts (to some extent) while generalization remains more vulnerable.

• Building on the complementary role of data shifts in MU, we propose a proactive, input-agnostic data perturbation strategy termed the *forget vector*, optimized specifically to facilitate unlearning.

• We demonstrate the effectiveness of *forget vector arithmetic* by using precomputed *class-wise* forget vectors to generate new vectors that effectively eliminate the influence of specific data subsets in image classification models, *e.g.*, in the scenario of *random* data forgetting.

• We conduct extensive experiments on MU for image classification, providing both quantitative and qualitative analyses to demonstrate the competitiveness of the forget vector compared to model-based MU methods.

## 2 RELATED WORK

**MU in Vision.** Machine unlearning (MU) in vision has gained significant attention due to the increasing need for privacy preservation, copyright protection, and ethical data removal in machine learning models. Recent studies (Gupta et al., 2021; Pan et al., 2022; Di et al., 2022; Zhang et al., 2024c; Liu et al., 2024a) in this area have primarily focused on two main applications: image classification and image generation.

In **image classification**, MU methods have explored various ways to erase certain classes or images from models (Golatkar et al., 2020; 2021; Thudi et al., 2022a; Chen et al., 2024; Liu et al., 2024a; Pochinkov & Schoots, 2024). Specifically, fine-tuning-based methods update the model incrementally on a modified dataset without the unwanted data points (Warnecke et al., 2021; Perifanis et al., 2024). Gradient ascent-based approaches attempt to reverse the impact of unwanted data by applying gradient ascent to model parameters (Thudi et al., 2022a; Chen et al., 2024). Influence-based unlearning

methods estimate and negate the effect of specific data points on model predictions and parameters to achieve unlearning (Golatkar et al., 2020; 2021). Another line explores the relationship between MU and model pruning, suggesting that model sparsity can help to bridge the gap between approximate and exact unlearning, reducing the need for complex parameter updates (Liu et al., 2024a).

For **image generation**, MU techniques (Fan et al., 2023; Zhang et al., 2024c) have been proposed to prevent models from generating unwanted or harmful content while retaining high-quality outputs. For example, weight saliency methods (Fan et al., 2023) guide MU by identifying and selectively altering model parameters to eliminate specific content generation. Beyond vision, MU has been applied to other domains, with notable efforts in natural language processing (Wang et al., 2023a; Shi et al., 2024; Wang et al., 2024; Liu et al., 2024b), graph-based data (Li et al., 2024; Dong et al., 2024), and time-series data (Du et al., 2019). However, most existing MU methods are model-based, requiring updates to model parameters and consequently incurring high computational costs.

**Input-based Model Adaptation.** This approach aims to modify or repurpose pre-trained models for new tasks or specific objectives without the need for full retraining. It is particularly beneficial for reducing computational costs and leveraging existing knowledge within models. Key techniques in model adaptation include: Visual prompting (Jia et al., 2022; Wang et al., 2023b; Liu et al., 2023; Hossain et al., 2024; Zhang et al., 2024a) maintains the pre-trained model's parameters fixed and adapts the input to enable the model to perform different tasks. For example, introducing trainable parameters in the input space while keeping the model backbone frozen can achieve comparable results with reduced computational overhead. Model reprogramming (Elsayed et al., 2018; Tsai et al., 2020; Yang et al., 2021; Melnyk et al., 2023) involves keeping a pre-trained model unchanged while modifying its inputs to adapt the model for new tasks. For example, adversarial perturbations can be applied to inputs at test time, allowing the model to perform a specific task dictated by the perturbations, even if that task was not originally intended for the model. Feature-based domain adaptation (Tahmoresnezhad & Hashemi, 2017; Wang et al., 2018) applies transformations or mapping techniques to the input data, aligning the feature distributions between the source and target domains while keeping the model unchanged.

## 3 PRELIMINARIES ON MU AND PROBLEM STATEMENT

**Formulation of MU.** In this work, we focus on the problem of MU for image classification. Let $\mathcal{D} = \{\mathbf{x}_i, y_i\}_{i=1}^{N}$ represent a training set with $N$ examples, where $\mathbf{x}_i$ denotes the $i$th image data, and $y_i$ denotes its corresponding class label. Following the classic MU setting (Golatkar et al., 2020; Fan et al., 2023; Kurmanji et al., 2024a;b), we introduce a *forget set* $\mathcal{D}_{\mathrm{f}} \subseteq \mathcal{D}$, which specifies the training samples targeted for unlearning. Accordingly, the complement of $\mathcal{D}_{\mathrm{f}}$ is the *retain set*, *i.e.*, $\mathcal{D}_{\mathrm{r}} = \mathcal{D} \setminus \mathcal{D}_{\mathrm{f}}$. The *goal of MU* is to efficiently and effectively eliminate the influence of $\mathcal{D}_{\mathrm{f}}$ on an already-trained model $\boldsymbol{\theta}_{\mathrm{o}}$, so that the performance of the post-unlearning model closely approximates that of a model retrained from scratch on $\mathcal{D}_{\mathrm{r}}$ (*i.e.*, excluding the impact of $\mathcal{D}_{\mathrm{f}}$ from scratch). Therefore, such a retraining method (referred to as *Retrain*) is typically considered as the gold standard of MU (Thudi et al., 2022a; Jia et al., 2023). However, since Retrain is computationally intensive, most popular MU approaches instead address an unlearning optimization problem using the forget and retain sets to update the model parameters $\boldsymbol{\theta}$, starting from the originally pre-trained model $\boldsymbol{\theta}_{\mathrm{o}}$. This yields the following optimization problem for MU:

$$\operatorname*{minimize}_{\boldsymbol{\theta}} \quad \ell_{\mathrm{MU}}(\boldsymbol{\theta}; \mathcal{D}_{\mathrm{f}}, \mathcal{D}_{\mathrm{r}}), \tag{1}$$

with the initialization $\boldsymbol{\theta} = \boldsymbol{\theta}_{\mathrm{o}}$. In (1), $\ell_{\mathrm{MU}}$ represents an appropriate unlearning loss function that may depend on $\mathcal{D}_{\mathrm{f}}$ and/or $\mathcal{D}_{\mathrm{r}}$, as will be detailed when introducing specific unlearning methods. In the context of MU for image classification (Golatkar et al., 2020; Fan et al., 2023), the specification of the forget set $\mathcal{D}_{\mathrm{f}}$ leads to two unlearning scenarios: *class-wise forgetting*, where $\mathcal{D}_{\mathrm{f}}$ consists of a subset focused on a specific image class targeted for unlearning, and *random data forgetting*, where $\mathcal{D}_{\mathrm{f}}$ is a randomly selected subset of images across all classes.

**Model-based MU Methods and Evaluation.** The formulation in (1) represents the predominant MU solution in the literature, focusing on modifying model weights and/or architectural components to achieve the unlearning objective. In what follows, we introduce several representative MU approaches that serve as approximations to Retrain. (a) Fine-tuning **(FT)** (Warnecke et al., 2021): This approach treats the MU problem as a continual learning task, defining the unlearning objective $\ell_{\mathrm{MU}}$ as a training objective that fine-tunes $\boldsymbol{\theta}_{\mathrm{o}}$ over $\mathcal{D}_{\mathrm{r}}$ to induce catastrophic forgetting of $\mathcal{D}_{\mathrm{f}}$. (b) Random labeling (**RL**) (Golatkar et al., 2020): This approach specifies the unlearning objective $\ell_{\mathrm{MU}}$

by assigning random labels or features to the data in $\mathcal{D}_{\mathrm{f}}$, thereby enforcing model forgetting. (c) Gradient ascent (**GA**) (Thudi et al., 2022a): This approach employs the negative of the FT loss to reverse the training impact associated with the data in $\mathcal{D}_{\mathrm{f}}$. (d) **Localization**-informed unlearning (Jia et al., 2023; Fan et al., 2023): This method identifies a subset of model weights critical to the unlearning task (*e.g.*, through model sparsity (Jia et al., 2023) or gradient saliency (Fan et al., 2023)) and incorporates this weight localization as a prior to solve the unlearning problem in (1).

Given an unlearned model (denoted as $\boldsymbol{\theta}_{\mathrm{u}}$) after solving (1), unlearning performance is evaluated in two main areas: *unlearning effectiveness*, which measures whether the target data/information has been successfully removed, and *utility retention*, which assesses whether unlearning has preserved the model's classification ability on unaffected data. Following the evaluation pipeline in Jia et al. (2023), unlearning effectiveness is quantified by two metrics: unlearning accuracy (**UA**), defined as $1-$the model's accuracy on $\mathcal{D}_{\mathrm{f}}$ (higher UA indicates better unlearning), and membership inference attack performance on $\mathcal{D}_{\mathrm{f}}$, termed **MIA-Efficacy**, where higher prediction accuracy on non-training samples indicates better unlearning (see Appendix A). Utility retention is measured by retain accuracy (**RA**), reflecting the model's accuracy on $\mathcal{D}_{\mathrm{r}}$, and testing accuracy (**TA**), which is the accuracy on the original test set. Notably, TA is assessed on the entire original test set, except in the case of class-wise forgetting, where test samples from the forgotten class are excluded from evaluation.

**Data-based MU Design: The Forget Vector Problem.** While previous MU methods can be unified within the framework of (1) by varying the unlearning loss $\ell_{\mathrm{MU}}$ and weight localization priors, recent advancements in input data-based model adaptation, such as visual prompting (Bahng et al., 2022a; Chen et al., 2023a) and model reprograming (Chen, 2024; Elsayed et al., 2018), suggest an alternative approach to MU. This strategy inspires us to design data-based prompting (implemented through universal input perturbations) to achieve unlearning without modifying the model itself. We refer to this input perturbation vector, designed specifically for MU, as the **forget vector**. To be more specific, let $\boldsymbol{\delta}$ represent the data-agnostic input perturbations to be designed. The problem of constructing a forget vector for MU can be formulated as

$$\underset{\boldsymbol{\delta}}{\operatorname{minimize}} \quad \ell_{\mathrm{MU}}(\boldsymbol{\delta}; \boldsymbol{\theta}_{\mathrm{o}}, \mathcal{D}_{\mathrm{f}}, \mathcal{D}_{\mathrm{r}}), \tag{2}$$

where $\boldsymbol{\delta}$ is the perturbation variable, applied linearly to the forget and retain samples as $\mathbf{x}' := \mathbf{x} + \boldsymbol{\delta}$ for $\mathbf{x}$ in $\mathcal{D}_{\mathrm{f}}$ and $\mathcal{D}_{\mathrm{r}}$, similar to visual prompting (Bahng et al., 2022a) and adversarial examples (Goodfellow et al., 2014). In practice, since the model remains unchanged, the unlearner can compute the forget vector based on the forget request (forget set) and append it to model inputs to process user-initiated unlearning requests. In this work, we do not consider counter-unlearning adversaries that intentionally negate the effect of the forget vector. We will detail the unlearning objective function required for designing the forget vector in our later method sections.

Based on (2), we are motivated to explore two research questions: **(Q1)** How do "perturbations" applied to forget data affect unlearning performance? **(Q2)** How can we effectively design the forget vector $\boldsymbol{\delta}$ to solve problem (2)? These two questions are interconnected: the answer to (Q1) offers a sensitivity analysis of MU to data shifts within the forget set, guiding how the specific shift induced by the forget vector can be optimized for effective unlearning in (Q2). Therefore, the following Secs. 4-5 address (Q1) and (Q2) in sequence. For (Q1), the next section analyzes performance through an evaluation lens on a given unlearned model, using data perturbations applied via standard data augmentation operations or adversarial perturbations.

## 4  GENERALIZATION OF MU TO FORGET DATA SHIFTS

Before designing the forget vector as formulated in (2), we examine the sensitivity of existing model-based unlearning approaches to external perturbations applied to forget data. Such a perturbation-based or out-of-distribution (OOD) generalization analysis of MU has not been explored in the literature. Our rationale is that if conventional MU approaches demonstrate robustness to these external forget data perturbations post unlearning, then enhancing MU with a forget vector could become a seamless process, as a proactive design of such a vector would likely yield effective results.

**Post-unlearning Forget Data Perturbations.** Given an unlearned model ($\boldsymbol{\theta}_{\mathrm{u}}$) after solving (1), we examine two types of shifts in forget data: standard data corruptions used in the evaluation of OOD generalization (Hendrycks & Dietterich, 2019; Hendrycks et al., 2021) and worst-case perturbations generated by adversarial attacks (Goodfellow et al., 2014; Madry et al., 2017). **(a) Data Corruptions**. Following the OOD generalization evaluation approach in image classification (Hendrycks & Dietterich, 2019), we consider four data corruptions : noise, blur, weather, and digital.

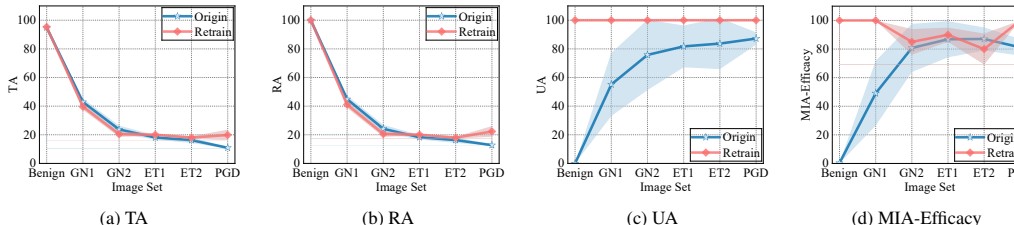

Figure 2: The performance of class-wise forgetting on (ResNet-18, CIFAR-10) using the unlearning method *Retrain* vs. the (pre-unlearning) original model performance (Origin), evaluated on both benign evaluation sets (Benign) and perturbed sets, which include (1) Gaussian noise (GN) with a standard deviation of 0.08 (termed GN1), (2) GN with a standard deviation of 0.2 (termed GN2), (3) Elastic transformation (ET) with parameters (488, 170.8, 24.4) regarding intensity, smoothing, and offset (termed ET1), (4) ET with parameters (488, 19.52, 48.8) (termed ET2), and (5) adversarial perturbations from a 7-step PGD attack with strength $\epsilon = 8/255$. The unlearning performance metrics are reported as (a) TA (testing accuracy), (b) RA (retain accuracy), (c) UA (unlearning accuracy), and (d) MIA-Efficacy, as defined in Sec. 3. The average performance is reported over 10 independent trials, where each trial focuses on forgetting one specific class from CIFAR-10. Shaded regions indicate the performance variance.

Each type of corruption includes five levels of severity, with higher levels denoting increased noise intensity. Among these, we select zero-mean Gaussian noise (GN) and Elastic transformations (ET) as the primary corruption types to evaluate MU robustness against shifts in forget data. Our rationale is that Gaussian noise yields small pixel-wise perturbations (similar to adversarial perturbations introduced later) and Elastic transformations stretch or contract small image regions. **(b) Adversarial perturbations.** An adversarial image is a benign image altered with carefully crafted, pixel-wise perturbations designed to mislead a classifier. In this work, we use the $\epsilon$-constrained $\ell_\infty$ norm-based $K$-step projected gradient descent (PGD) attack (Goodfellow et al., 2014; Madry et al., 2017) to generate adversarial examples via iterative projected gradient updates. The parameter $\epsilon > 0$ defines the radius of the $\ell_\infty$ norm of the perturbations, controlling their strength. And $K$ represents the number of PGD steps.

**Generalization of MU to Forget Data Perturbations.** Next, we apply the above data shift operations to the MU evaluation sets, namely, the forget, retain, and testing sets, and assess the unlearning performance of an unlearned model. **Fig. 2** displays the performance of the gold standard unlearning method, Retrain, against Gaussian noise at test time with standard deviations of 0.08 and 0.2 (Hendrycks & Dietterich, 2019), and two types of Elastic transformations with parameters (488, 170.8, 24.4) and (488, 19.52, 48.8) regarding intensity, smoothing and offset for moderate and high-intensity distortions (Hendrycks & Dietterich, 2019), as well as a 7-step PGD attack with perturbation strength $\epsilon = 8/255$ (Goodfellow et al., 2014). To ensure the feasibility of Retrain, we conduct the image classification task using ResNet-18 on the CIFAR-10 dataset.

As shown in Fig. 2-(a) and (b), model utility, measured by RA (retain accuracy) and TA (testing accuracy), decreases when external perturbations are applied to the evaluation sets compared to its original performance without perturbations. This is expected due to the generalization loss when evaluated on new, shifted data. More interestingly, Fig. 2-(c) and (d) show that unlearning effectiveness of Retrain, measured by UA (unlearning accuracy) and MIA-Efficacy, remains stable despite the presence of these perturbations on the forget set. This is because perturbations degrade prediction performance across evaluation sets, including the forget set. This is further evidenced by the increase in UA and MIA-Efficacy for the original model (without unlearning) when exposed to data perturbations. Above indicates that a reduction in performance on the forget set could translate into enhanced unlearning effectiveness on that set. In Appendix B, we provide additional evaluations of other approximate MU methods, including FT, RL, and GA, showing consistent performance.

The results above demonstrate that unlearning effectiveness is inherently preserved under external perturbations at no additional cost. However, balancing this with utility retention in the presence of perturbations remains challenging and desirable. Therefore, we need to carefully address the forget vector problem (2) to develop an input-based MU solution that enhances unlearning effectiveness without compromising model utility.

## 5 OPTIMIZATION FOR FORGET VECTORS

**Unlearning Objective Design of Forget Vectors.** Our design aims for the forget vector variable ($\delta$), when applied to the forget set ($\mathcal{D}_f$), to drive the given model's predictions ($\theta_o$) away from the correct labels. Conversely, when applied to the retain set ($\mathcal{D}_r$), the forget vector should minimally

affect correct predictions. The first forget objective aligns with adversarial attack design, aiming to mislead the model's predictions in the presence of the perturbation $\boldsymbol{\delta}$. The second retain objective acts as a utility regularization, suppressing the unlearning effect of the perturbation when applied to data samples not targeted for unlearning (retain samples).

To implement the forget objective (denoted by $\ell_{\mathrm{f}}$), we draw inspiration from the C&W untargeted attack loss (Carlini & Wagner, 2017). This is given by a margin loss, designed to remain actively minimizing when the top prediction matches the correct label, ensuring that optimization continues until predictions are shifted to an incorrect label, thereby achieving unlearning. This can be cast as

$$\ell_{\mathrm{f}}(\boldsymbol{\delta}; \boldsymbol{\theta}_{\mathrm{o}}, \mathcal{D}_{\mathrm{f}}) = \mathbb{E}_{(\mathbf{x}, y) \in \mathcal{D}_{\mathrm{f}}} \max\{f_{\boldsymbol{\theta}_{\mathrm{o}}, y}(\mathbf{x} + \boldsymbol{\delta}) - \max_{k:\, k \neq y} f_{\boldsymbol{\theta}_{\mathrm{o}}, k}(\mathbf{x} + \boldsymbol{\delta}), -\tau\}, \tag{3}$$

where $(\mathbf{x}, y) \in \mathcal{D}_{\mathrm{f}}$ denotes a forget sample with $y$ being the prediction label of $\mathbf{x}$, $\mathbf{x} + \boldsymbol{\delta}$ is the perturbed sample, $f_{\boldsymbol{\theta}_{\mathrm{o}}, k}(\mathbf{x})$ denotes the prediction logit (before softmax) of the model $\boldsymbol{\theta}_{\mathrm{o}}$ for class $k$ under the input $\mathbf{x}$, and $\tau \geq 0$ is a margin threshold that controls the unlearning strength. The rationale behind (3) is that minimizing it ensures convergence to the negative margin $f_y(\mathbf{x} + \boldsymbol{\delta}) - \max_{k \neq y} f_k(\mathbf{x} + \boldsymbol{\delta}) \to -\tau \leq 0$. Thus, the forget vector $\boldsymbol{\delta}$ enforces unlearning on $\boldsymbol{\theta}$ for $\mathbf{x}$ by making the incorrect prediction $(k \neq y)$ have a higher confidence than the original correct prediction $y$. On the other hand, once the margin becomes negative (indicating that the prediction label has been flipped), the forget objective $\ell_{\mathrm{f}}$ automatically terminates, allowing a balance with the retain objective, which will be introduced later. In our experiments, we find that the forget objective is robust to variations in the *nonnegative* margin parameter $\tau$ (see Appendix C). A larger $\tau$ value imposes a stricter unlearning requirement by increasing the logit distance from the correct label. For example, we set $\tau = 1$ in our experiments.

Next, we regularize the forget objective (3) with the retain objective, defined as the cross-entropy loss ($\ell_{\mathrm{CE}}$) over the retain set $\mathcal{D}_{\mathrm{r}}$, along with the $\ell_2$ norm of $\boldsymbol{\delta}$ to ensure minimal perturbation required to achieve both the forget and retain objectives. This yields the full unlearning objective in (2):

$$\ell_{\mathrm{MU}}(\boldsymbol{\delta}; \boldsymbol{\theta}_{\mathrm{o}}, \mathcal{D}_{\mathrm{f}}, \mathcal{D}_{\mathrm{r}}) = \ell_{\mathrm{f}}(\boldsymbol{\delta}; \boldsymbol{\theta}_{\mathrm{o}}, \mathcal{D}_{\mathrm{f}}) + \lambda_1 \ell_{\mathrm{CE}}(\boldsymbol{\delta}; \boldsymbol{\theta}_{\mathrm{o}}, \mathcal{D}_{\mathrm{r}}) + \lambda_2 \|\boldsymbol{\delta}\|_2^2, \tag{4}$$

where $\lambda_1 > 0$ and $\lambda_2 > 0$ are the regularization parameters, and $\ell_{\mathrm{CE}}(\boldsymbol{\delta}; \boldsymbol{\theta}_{\mathrm{o}}, \mathcal{D}_{\mathrm{r}})$ denotes the CE loss of the model $\boldsymbol{\theta}_{\mathrm{o}}$ over the perturbed retain set $\{(\mathbf{x} + \boldsymbol{\delta}, y)\}_{(\mathbf{x}, y) \in \mathcal{D}_{\mathrm{r}}}$. Integrating (4) into (3), we can then apply stochastic gradient descent (SGD) (Amari, 1993) to optimize the forget vector variable $\boldsymbol{\delta}$.

**Compositional Unlearning via Forget Vector Arithmetic.** A forget vector defines an unlearning direction in the input space to guide the unlearning process. We explore whether a *new* unlearning direction can be efficiently constructed by interpolating from existing precomputed forget vectors, such as class-wise forget vectors obtained by solving (4) with $\mathcal{D}_{\mathrm{f}}$ defined as each class's training set. This approach is analogous to the concept of task vectors in weight space for model editing (Ilharco et al., 2022). However, to the best of our knowledge, *input-based* task vector arithmetic has not yet been explored in the literature. If forget vectors can be modified and combined using arithmetic operations, such as negation and addition, we can dynamically adjust a model's unlearning behavior without re-solving the optimization problem (4) or any other model-based unlearning problem in (1). We refer to this new unlearning paradigm as *compositional unlearning*, where precomputed class-wise forget vectors can be efficiently combined to generate a new forget vector for each deletion request in the context of random data forgetting.

Let $\boldsymbol{\delta}_k$ denote the forget vector used for unlearning data points of class $k$. Given the set of forget vectors $\{\boldsymbol{\delta}_k\}_{k=1}^K$ for all $K$ classes, we obtain these vectors by solving (4) with $\mathcal{D}_{\mathrm{f}}$ defined as each class's training set, respectively. The forget vector for compositional unlearning is given by

$$\boldsymbol{\delta}(\mathbf{w}) := \sum_{k=1}^{K} (w_k \boldsymbol{\delta}_k), \tag{5}$$

where $\mathbf{w} = [w_1, \ldots w_K]^K$ are the linear combination coefficients to be optimized, which determine the forget vector arithmetic. To determine $\mathbf{w}$, we can minimize (4) with the optimization restricted to the coefficients $\mathbf{w}$. Instead of penalizing the $\ell_2$ norm of the forget vector, we penalize the $\ell_2$ norm of $\mathbf{w}$ to prevent excessive pixel perturbation. This modifies (4) to the problem $\min_{\mathbf{w}} \ell_{\mathrm{f}}(\boldsymbol{\delta}(\mathbf{w}); \boldsymbol{\theta}_{\mathrm{o}}, \mathcal{D}_{\mathrm{f}}) + \lambda_1 \ell_{\mathrm{CE}}(\boldsymbol{\delta}(\mathbf{w}); \boldsymbol{\theta}_{\mathrm{o}}, \mathcal{D}_{\mathrm{r}}) + \lambda_2 \|\mathbf{w}\|_2^2$. As will be shown later, random data forgetting can be achieved through class-wise forget vector arithmetic (5) by applying the compositional scheme defined by $\mathbf{w}$.

To illustrate the effectiveness of forget vector arithmetic, **Fig. 3** shows preliminary results of combining two class-wise forget vectors ($\boldsymbol{\delta}_1$ and $\boldsymbol{\delta}_2$) using a simple scheme $\boldsymbol{\delta}(\mathbf{w}) = w_1 \boldsymbol{\delta}_1 + w_2 \boldsymbol{\delta}_2$ on (CIFAR-10,

ResNet-18), when forgetting a randomly selected 10% of training points from class "automobile" and class "bird" refering to $w_1$ and $w_2$. Rather than optimizing $\mathbf{w}$, we adjust $w_1$ and $w_2$ from $-0.2$ to $0.2$ to observe how the performance gap relative to Retrain varies. This evaluation includes

the UA (unlearning accuracy) gap on the selected forget data, the RA (retain accuracy) gap on the remaining data, and the average gap across these two metrics. As expected, Fig. 3-(a) and (b) shows a trade-off among these two metrics, where weight configurations that achieve a low UA gap may result in a higher RA gap, and vice versa. Additionally, Fig. 3-(c) shows that moderate weight values of $w_1$ and $w_2$ ($-0.1 \leq w_1 \leq 0.1$ and $-0.1 \leq w_2 \leq 0.1$ )

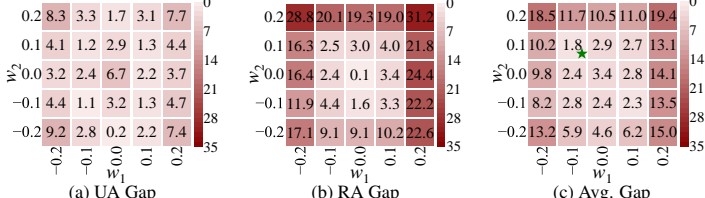

(a) UA Gap      (b) RA Gap      (c) Avg. Gap

Figure 3: The performance gap relative to Retrain for class-wise forget vector arithmetic (based on classes "automobile" and "bird") across different combination coefficients $w_1$ and $w_2$, when unlearning a randomly selected 10% of training points from these two classes of CIFAR-10. Each cell displays the gap (%) relative to Retrain at a specific weight combination, where a lower value indicates a closer performance to Retrain given a metric. A green star (★) denotes the selected weight combination scheme ($w_1$ and $w_2$) that achieves the smallest performance gap relative to Retrain, averaged over both UA Gap and RA Gap.

tend to yield a more balanced average performance, maintaining relatively low gaps across two metrics. A favored weighting scheme can be identified at $w_1 = -0.1$ and $w_2 = 0.1$ as marked by the green star (★), validating the feasibility of arbitrary random data forgetting using our proposed compositional unlearning via the forget vector arithmetic approach.

## 6 EXPERIMENTS

### 6.1 EXPERIMENT SETUPS

**Datasets and Models.** We focus on MU for image classification, using two datasets: CIFAR-10 (Krizhevsky et al., 2009) and ImageNet-10, a 10-class subset of the original ImageNet (Deng et al., 2009), for ease of implementation of Retrain (exact unlearning) over ImageNet images as (Tao et al., 2021). For these tasks, we use three image classifiers: ResNet-18 (He et al., 2016) for CIFAR-10, VGG-16 (Simonyan & Zisserman, 2014) and ViT-Base (Dosovitskiy et al., 2020) for ImageNet.

**Unlearning Baselines and Evaluations.** In the context of MU for image classification, we consider two scenarios: *class-wise forgetting* and *random data forgetting*. In class-wise forgetting, training data from an image class are designated for unlearning, while in random data forgetting, a subset of all-class training points is randomly selected as the forget set, with a specified forget ratio of 10%. We consider 8 MU baseline methods, including ① FT (Warnecke et al., 2021), ② RL (Golatkar et al., 2020), ③ GA (Thudi et al., 2022a), ④ NegGrad+ (Kurmanji et al., 2024a), ⑤ SalUn (Fan et al., 2023), ⑥ SCRUB (Kurmanji et al., 2024b), ⑦ Class-F (Kodge et al., 2024), and ⑧ SSD (Foster et al., 2024), where Class-F is only designed for class-wise forgetting.

As described in Sec. 3, unlearning effectiveness is measured using UA (unlearning accuracy) and MIA-Efficacy, while model utility post-unlearning is assessed by RA (retain accuracy) and TA (testing accuracy); For all metrics, being closer to Retrain indicates better performance. It is also worth noting that all existing model-based MU baseline methods ①-⑧ are evaluated on non-perturbed evaluation sets. However, when using our proposed data-based forget vector solution, we need to apply the forget vector to the evaluation sets (including the forget set, retain set, and testing set) in order to assess unlearning effectiveness and utility retention. This evaluation remains fair, as it aligns with the same objective of forgetting targeted data. The key distinction is that the forget vector operates at the input level, whereas model-based MU baselines achieve unlearning by modifying model weights. To quantify the performance gap with Retrain, we compare each unlearning baseline and our proposal against this exact unlearning method across all metrics. We report an averaged assessment, termed *Averaging (Avg.) Gap*. Unless specified otherwise, all the main experiments (whether class-wise or random data forgetting) are conducted over 10 random trials, with mean performance reported.

**Implementation Details of Our Proposal.** To solve the forget vector problem (2) with the proposed unlearning objective in (4), we set the retain loss regularization parameter $\lambda_1$ as follows: 3 for CIFAR-10, 5 for ImageNet-10 with VGG-16, and 7 for ImageNet-10 with ViT-Base in class-wise forgetting. For random data forgetting, we set $\lambda_1$ to 1. The $\ell_2$-norm regularization parameter is set to $\lambda_2 = 1$. These hyperparameters are determined through a grid search over the range $[0, 10]$. To optimize

Table 1: Performance overview of various MU methods for image classification under two unlearning scenarios on CIFAR-10 using ResNet-18 and ImageNet-10 using ViT-Base. Since Class-F (Kodge et al., 2024) is specifically designed for class-wise forgetting, its results do not apply to random data forgetting scenarios (n/a). Other results are reported in the format $a_{\pm b}$, where $a$ is the mean and $b$ denotes standard deviation $b$ over 10 independent trials. The performance gap against Retrain is indicated in (•), where a lower value is better. ↑ (or ↓) indicates that a higher (or lower) value is better. The best performance for each metric is highlighted in green, while the second-best performance is highlighted in red.

| MU Method | UA↑ | MIA-Efficacy↑ | RA↑ | TA↑ | Avg.Gap↓ | UA↑ | MIA-Efficacy↑ | RA↑ | TA↑ | Avg.Gap↓ |
|---|---|---|---|---|---|---|---|---|---|---|
| | Class-wise Forgetting, CIFAR-10, ResNet-18 | | | | | Random Data Forgetting, CIFAR-10, ResNet-18 | | | | |
| Retrain | 100.00±0.00(0.00) | 100.00±0.00(0.00) | 99.91±0.03(0.00) | 94.92±0.15(0.00) | 0.00 | 5.50±0.16(0.00) | 11.57±0.47(0.00) | 99.88±0.05(0.00) | 94.24±0.19(0.00) | 0.00 |
| FT | 5.27±0.73(94.73) | 51.49±5.07(48.51) | 100.0±0.00(0.09) | 95.03±0.07(0.11) | 35.86 | 0.03±0.03(5.47) | 0.75±0.09(10.82) | 99.98±0.02(0.10) | 94.45±0.14(0.21) | 4.15 |
| RL | 18.87±7.34(81.13) | 98.94±0.79(1.06) | 99.98±0.0(0.07) | 94.51±0.12(0.41) | 20.67 | 0.52±0.24(4.98) | 3.13±0.55(8.44) | 99.85±0.07(0.03) | 93.88±0.20(0.36) | 3.45 |
| GA | 71.45±0.35(28.55) | 81.7±0.22(18.30) | 98.62±0.04(1.29) | 92.34±0.02(2.58) | 12.68 | 1.56±0.38(3.94) | 2.88±3.44(8.69) | 98.67±2.74(1.21) | 92.84±2.59(1.40) | 3.81 |
| NegGrad+ | 91.78±14.66(8.22) | 95.81±7.38(4.19) | 98.35±1.22(1.56) | 92.62±1.34(2.30) | 4.07 | 0.97±1.08(4.53) | 2.74±2.16(8.83) | 99.42±0.87(0.46) | 93.38±1.13(0.86) | 3.67 |
| SalUn | 96.35±2.14(3.65) | 98.64±0.03(1.36) | 98.75±0.18(1.16) | 92.34±1.54(2.58) | 2.19 | 1.73±0.25(3.77) | 6.25±1.21(5.32) | 99.24±0.09(0.64) | 91.03±0.14(3.21) | 3.23 |
| SCRUB | 93.45±2.33(6.55) | 96.38±1.72(3.62) | 99.95±0.0(0.04) | 94.56±0.07(0.36) | 2.64 | 0.61±0.31(4.89) | 3.69±0.54(7.88) | 99.76±0.18(0.12) | 93.91±0.19(0.33) | 3.31 |
| Class-F | 90.18±1.20(9.82) | 86.15±2.67(13.85) | 91.25±0.05(8.66) | 85.45±0.19(9.47) | 10.45 | n/a | n/a | n/a | n/a | n/a |
| SSD | 96.05±0.45(3.95) | 98.00±0.00(2.00) | 97.77±0.20(2.14) | 92.23±0.58(2.69) | 2.70 | 5.54±0.00(0.04) | 7.80±0.06(3.77) | 94.86±0.00(5.02) | 88.28±0.00(5.96) | 3.70 |
| Ours | 97.88±0.27(2.12) | 99.60±0.15(0.40) | 97.25±0.24(2.66) | 90.90±0.32(4.02) | 2.30 | 2.61±0.49(2.89) | 8.26±1.17(3.00) | 97.33±0.47(2.55) | 90.97±0.38(3.27) | 2.92 |
| | Class-wise Forgetting, ImageNet-10, ViT-Base | | | | | Random Data Forgetting, ImageNet-10, ViT-Base | | | | |
| Retrain | 100±0.00(0.00) | 100.00±0.00(0.00) | 99.97±0.03(0.00) | 99.85±0.01(0.00) | 0.00 | 1.41±0.06(0.00) | 93.57±0.00(0.00) | 99.07±0.01(0.00) | 99.27±0.01(0.00) | 0.00 |
| FT | 42.79±7.51(57.21) | 40.78±10.68(59.22) | 99.96±0.01(0.01) | 99.61±0.10(0.24) | 29.17 | 1.38±0.16(0.03) | 96.40±0.31(2.83) | 99.60±0.09(0.53) | 99.10±0.30(0.17) | 0.89 |
| RL | 88.15±1.62(11.85) | 96.50±1.50(3.50) | 99.93±0.02(0.04) | 99.89±0.11(0.04) | 3.84 | 2.62±0.40(1.21) | 93.75±0.02(0.18) | 98.38±0.18(0.69) | 95.05±0.26(4.20) | 1.57 |
| GA | 28.05±7.05(71.95) | 58.73±5.83(41.27) | 99.97±0.01(0.00) | 99.70±0.10(0.15) | 28.34 | 0.82±0.16(0.59) | 96.77±1.10(3.20) | 99.60±0.09(0.53) | 99.53±0.09(0.26) | 1.15 |
| NegGrad+ | 10.90±2.87(89.10) | 79.30±4.58(20.70) | 99.98±0.00(0.01) | 99.78±0.00(0.07) | 27.46 | 1.97±0.72(0.56) | 95.48±1.67(1.91) | 98.09±0.94(0.98) | 98.08±0.75(1.19) | 1.16 |
| SalUn | 93.27±1.50(6.73) | 94.00±1.00(6.00) | 98.22±0.75(1.75) | 98.00±0.00(1.85) | 4.08 | 0.67±0.19(0.74) | 95.80±0.04(2.23) | 99.65±0.07(0.58) | 98.27±0.09(0.00) | 1.14 |
| SCRUB | 99.10±0.20(0.90) | 95.67±1.77(4.33) | 98.90±0.10(1.07) | 99.33±0.31(0.52) | 1.71 | 0.85±0.24(0.56) | 95.90±0.95(2.33) | 99.58±0.23(0.51) | 99.20±0.14(0.07) | 0.87 |
| Class-F | 28.62±7.85(71.38) | 55.10±2.30(44.90) | 77.50±1.62(22.47) | 75.22±0.56(24.63) | 40.85 | n/a | n/a | n/a | n/a | n/a |
| SSD | 90.35±1.65(9.65) | 60.15±1.85(39.65) | 98.43±0.55(1.54) | 98.33±0.56(1.52) | 13.14 | 1.12±0.40(0.29) | 94.15±0.35(0.58) | 98.97±0.00(0.10) | 99.40±0.00(0.13) | 0.28 |
| Ours | 95.92±0.27(4.08) | 99.40±0.14(0.60) | 99.13±0.04(0.84) | 99.26±0.28(0.59) | 1.53 | 1.08±0.39(0.33) | 91.40±1.30(2.17) | 98.97±0.35(0.10) | 99.90±0.50(0.17) | 0.69 |

(2), we use stochastic gradient descent (SGD) (Amari, 1993) with a momentum factor of 0.9 and an exponential learning rate scheduler, decaying at a rate of 0.9 per iteration. Additionally, the batch size is set to 256, with a maximum of 40 optimization iterations for both two datasets. To solve the compositional unlearning problem (5), we use a similar setup, setting both $\lambda_1$ and $\lambda_2$ to 1.

## 6.2 EXPERIMENT RESULTS

**Overview Performance of Forget Vector.** In Tab. 1, we compare the performance of our forget vector approach with other model-based MU methods across the metrics: UA, RA, TA, MIA-Efficacy, and Avg. Gap vs. Retrain. We highlight three key observations below. **First**, in terms of unlearning effectiveness (UA and MIA-Efficacy), the data perturbation-based forget vector demonstrates highly competitive performance compared to model update-based MU baselines, mostly ranking among the top two methods with the smallest performance gap relative to Retrain (as evidenced by Avg. Gap). The advantage of the forget vector is particularly evident in MIA-Efficacy, where it usually achieves the closest results to Retrain. **Second**, in terms of model utility post-unlearning (RA and TA), the forget vector generally leads to a larger performance drop than other methods. This is not surprising, as the forget vector is achieved through data perturbations. However, considering the gain in unlearning effectiveness, the Avg. Gap with Retrain shows that the forget vector remains competitive, ranking among the top two unlearning methods. **Third**, unlearning methods (including Retrain) do not exhibit the same level of distinctiveness in random data forgetting as it does in class-wise forgetting. This is because in random data forgetting, the retain data could have sufficiently represented the distribution of the forget data, making it more challenging for MIA to distinguish forgotten samples from retained ones. Besides, the corresponding results of various MU methods on ImageNet-10 using VGG-16 can be found in Appendix D.

**Transferability of Forget Vector to Unseen Forget Data.** Conventionally, unlearning effectiveness is typically measured on the original forget set ($\mathcal{D}_f$). However, with the data perturbation-based forget vector, it is also interesting to investigate its unlearning transferability when applied to a new, previously unseen forget set (denoted as $\mathcal{D}_f'$) that share similarities with $\mathcal{D}_f$ and are equally appropriate for unlearning. In the context of class-wise forgetting, we consider $\mathcal{D}_f'$ using the testing data from the class targeted for unlearning, where unlearning performance should align closely with Retrain since the test-time data to forget share the same distribution with the training set. In the context of data forgetting, we obtain $\mathcal{D}_f'$ by applying the data corruption operation GN1 and ET1 used

Table 2: UA (%) of forget vector when transferred to unseen forget sets curated under 3 scenarios on (CIFAR-10, ResNet-18). The results are presented in the same format as Table 1.

| MU Method | $D'_f$ (Class-wise) from testing set | $D'_f$ (Random Data) perturbed by GN1 | $D'_f$ (Random Data) perturbed by ET1 |
|---|---|---|---|
| Retrain | 100.00±0.00(0.00) | 64.73±3.36(0.00) | 81.43±0.28(0.00) |
| FT | 21.44±1.11(78.56) | 56.75±1.40(7.98) | 81.79±0.18(0.36) |
| RL | 27.90±5.70(72.10) | 61.84±2.45(2.89) | 81.15±0.20(0.28) |
| GA | 73.95±0.60(26.05) | 55.20±2.59(9.53) | 82.17±0.76(1.28) |
| NegGrad+ | 93.86±10.93(6.14) | 57.81±1.76(6.92) | 81.73±0.65(0.30) |
| SalUn | 97.55±1.37(2.45) | 73.15±4.25(8.42) | 80.27±3.56(1.16) |
| SCRUB | 93.65±2.65(6.35) | 61.95±0.86(1.85) | 81.18±0.62(0.25) |
| Class-F | 89.14±5.13(10.66) | n/a | n/a |
| SSD | 98.70±0.03(1.30) | 80.34±0.01(15.91) | 84.70±0.02(3.27) |
| Ours | 98.26±0.35(1.74) | 78.32±1.03(13.59) | 85.03±0.91(3.60) |

in Fig. 2 to perturb $\mathcal{D}_f$ (last two columns of Tab. 2), where Retrain is no longer the gold standard as training data distribution excludes these shifts, allowing higher UA for better unlearning. As observed

in Tab. 2, the unlearning performance of the forget vector remains effective when evaluated on $\mathcal{D}'_f$. Among the model update-based unlearning baselines, current SOTA methods such as SCRUB, SalUn and SSD also demonstrate generalization to $\mathcal{D}'_f$, compared to simpler MU methods like FT, RL, and GA, which show lower transferability.

**Compositional Unlearning by Class-wise Forget Vectors.** Next, we demonstrate the effectiveness of compositional unlearning via forget vector arithmetic (termed CU-FV). Given pre-computed class-wise forget vectors, we apply their linear combination as defined in (5) to achieve random data forgetting. Tab. 3 compares the performance of CU-FV with Retrain (the exact unlearning method) and the direct forget vector approach (FV) applied to the targeted forget set.

Interestingly, we observe that CU-FV achieves the overall performance comparable to FV, as indicated by similar Avg. Gap values. Unlike FV, CU-FV optimizes only the class-wise coefficients in (5), resulting in a much smaller optimization space than FV. However, from UA and MIA-Efficacy metrics, we find that unlearning effectiveness is easier to maintain since unlearning typically targets a smaller subset of data points. In contrast, model utility (RA and TA) may decline more with CU-FV than with FV.

Table 3: Compositional unlearning on CIFAR-10 and ImageNet-10 for random data forgetting, where FV represents the original setting of forget vector that is directly learned based on a targeted forget set, and CU-FV denotes compositional unlearning achieved via pre-learned class-wise forget vector arithmetic. The results are presented in the same format as Table 1.

| Module | MU Method | UA↑ | MIA-Efficacy↑ | RA↑ | TA↑ | Avg.Gap↓ |
|---|---|---|---|---|---|---|
| CIFAR-10 ResNet-18 | Retrain | $5.50_{\pm0.16}(0.00)$ | $11.57_{\pm0.47}(0.00)$ | $94.24_{\pm0.19}(0.00)$ | $99.88_{\pm0.05}(0.00)$ | **0.00** |
| | FV | $2.61_{\pm0.49}(2.89)$ | $8.26_{\pm1.17}(3.00)$ | $90.97_{\pm0.38}(3.27)$ | $97.33_{\pm0.47}(2.55)$ | **2.92** |
| | CU-FV | $5.36_{\pm0.60}(0.14)$ | $9.76_{\pm0.91}(1.81)$ | $88.60_{\pm0.59}(5.64)$ | $94.93_{\pm0.64}(4.95)$ | **3.16** |
| ImageNet-10 VGG-16 | Retrain | $4.05_{\pm0.45}(0.00)$ | $6.60_{\pm1.07}(0.00)$ | $96.33_{\pm0.38}(0.00)$ | $99.48_{\pm0.07}(0.00)$ | **0.00** |
| | FV | $2.27_{\pm0.50}(1.78)$ | $6.13_{\pm1.40}(0.47)$ | $95.82_{\pm0.48}(0.51)$ | $98.29_{\pm0.32}(1.19)$ | **0.99** |
| | CU-FV | $2.27_{\pm1.18}(1.78)$ | $4.95_{\pm1.86}(1.65)$ | $91.41_{\pm1.25}(4.92)$ | $97.93_{\pm1.09}(1.55)$ | **2.47** |
| ImageNet-10 ViT-Base | Retrain | $1.41_{\pm0.06}(0.00)$ | $99.07_{\pm0.01}(0.00)$ | $93.57_{\pm0.00}(0.00)$ | $99.27_{\pm0.01}(0.00)$ | **0.00** |
| | FV | $1.08_{\pm0.39}(0.33)$ | $98.97_{\pm0.35}(0.10)$ | $91.40_{\pm1.30}(2.17)$ | $99.10_{\pm0.50}(0.17)$ | **0.69** |
| | CU-FV | $1.62_{\pm0.05}(0.21)$ | $98.55_{\pm0.14}(0.52)$ | $92.09_{\pm0.25}(1.48)$ | $98.50_{\pm0.01}(0.77)$ | **0.75** |

**Assessing Forget Vector via An Input Saliency Lens.** In Fig. 4, we explore the impact of the forget vector on unlearning and utility retention through input saliency map, using Grad-CAM (Gradient-weighted Class Activation Mapping) (Bengio et al., 2013). Using Grad-CAM, we visualize the salient pixels (*i.e.*, regions most influential to model prediction) for forget images under different unlearning scenarios: (1) the original model (without unlearning), (2) Retrain, (3) SCRUB-based unlearning, and (4) forget vector-based unlearning. In the first three scenarios, we obtain the input saliency maps on raw forget images without the addition of the forget vector, while

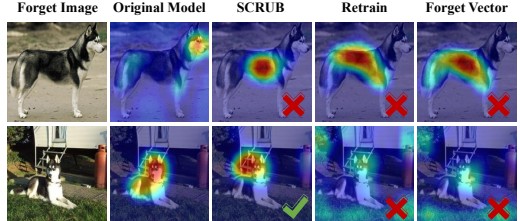

Figure 4: Gradient-based saliency map visualized via Grad-CAM for different MU methods against **forget images**. The highlighted areas (marked in red) indicate regions most influential to model prediction, and the red cross mark (✖) indicates that corresponding methods effectively unlearn the input forget images, while the check (✔) signifies the opposite.

in the last scenario, the input saliency is applied to images perturbed by the forget vector. As seen in Fig. 4 the forget vector and Retrain effectively unlearn the target input images, evident from the significant shifts in saliency regions compared to those in the original model. In contrast, the MU baseline SCRUB shows minimal saliency shifts, failing to adequately forget the target data in the last row. More visualization results on both forget and retain images can be found in Appendix G.

**Ablation Studies.** In Appendix E, we provide additional ablation studies on the sensitivity of the unlearning-retaining regularization parameter $\lambda_1$ in (4). Moreover, we perform the efficiency analysis of forget vector calculation and compositional unlearning in Appendix F.

## 7 CONCLUSIONS

In this paper, we introduce a novel, data-based approach to machine unlearning (MU) in image classification, termed the *forget vector*. Unlike traditional model-based MU methods that require retraining or fine-tuning model weights, our approach shows that input-agnostic data perturbations can effectively achieve unlearning objectives. Our method demonstrates competitive performance relative to model-based approximate unlearning techniques. Furthermore, we showcase the potential of compositional unlearning: new forget vectors for unseen tasks, such as unlearning arbitrary subsets across classes, can be generated through simple arithmetic operations, like linear combinations of class-specific forget vectors. Extensive experiments confirm the effectiveness and adaptability of our optimized forget vector. We refer readers to Appendix H-I for broader impacts, limitations, and details of LLM usage.

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

APPENDIX

# A  DETAILS OF MIA IMPLEMENTATION

To evaluate the effectiveness of the unlearning process, MIA is implemented following (Jia et al., 2023) using a prediction confidence-based attack method (Song & Mittal, 2021), which comprises a training phase and a testing phase in its computation. Specifically, a balanced dataset is first formed by sampling data points from retain set $\mathcal{D}_r$ and test set $\mathcal{D}_t$, ensuring the distinction from the forget set $\mathcal{D}_f$. Then, a MIA predictor is trained utilizing such balanced dataset. Thereafter, MIA-Efficacy can be calculated by applying the trained MIA predictor to the unlearned model $\theta_u$ on the forget set $\mathcal{D}_f$. Essentially, the goal is to determine how many samples in $\mathcal{D}_f$ can be accurately identified as non-training data with respect to $\theta_u$ by the MIA model. Formally, MIA-Efficacy is defined as follows,

$$\text{MIA-Efficacy} = N_{tn}/|\mathcal{D}_f|, \tag{6}$$

where $N_{tn}$ represents the total number of true negative samples in the forget set $\mathcal{D}_f$ predicted by the trained MIA model, *i.e.*, the number of forgetting samples classified as non-training examples.

# B  GENERALIZATION OF MU TO FORGET DATA SHIFTS

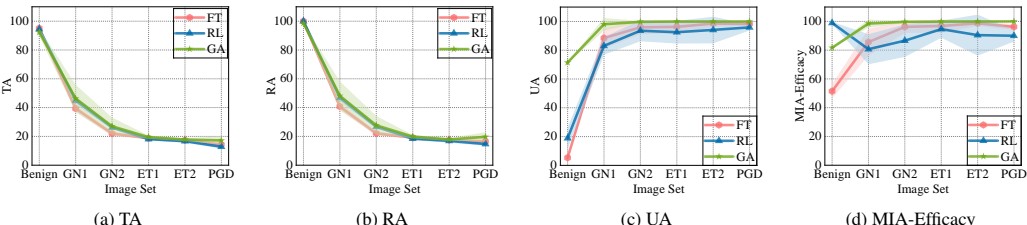

(a) TA      (b) RA      (c) UA      (d) MIA-Efficacy

Figure A1: The performance of class-wise forgetting on (ResNet-18, CIFAR-10) using the unlearning method FT, RL and GA, evaluated on both benign evaluation sets (Benign) and perturbed sets, which include (1) Gaussian noise (GN) with a standard deviation of 0.08 (termed GN1), (2) GN with a standard deviation of 0.2 (termed GN2), (3) Elastic transformation (ET) with parameters (488, 170.8, 24.4) regarding intensity, smoothing, and offset (termed ET1), (4) ET with parameters (488, 19.52, 48.8) (termed ET2), and (5) adversarial perturbations from a 7-step PGD attack with strength $\epsilon = 8/255$. The unlearning performance metrics are reported as (a) TA (testing accuracy), (b) RA (retain accuracy), (c) UA (unlearning accuracy), and (d) MIA-Efficacy, as defined in Sec.3 of main paper. The average performance is reported over 10 independent trials, where each trial focuses on forgetting one specific class from CIFAR-10. Shaded regions indicate the performance variance.

In **Fig. A1**, we provide additional evaluations of other approximate unlearning methods, including FT, RL, and GA against Gaussian noise at test time with standard deviations of 0.08 and 0.2 (Hendrycks & Dietterich, 2019), and two types of Elastic transformations with parameters (488, 170.8, 24.4) and (488, 19.52, 48.8) regarding intensity, smoothing and offset for moderate and high-intensity distortions (Hendrycks & Dietterich, 2019), as well as a 7-step PGD attack with perturbation strength $\epsilon = 8/255$ (Goodfellow et al., 2014). The experiments are conducted on the CIFAR-10 dataset using ResNet-18 for the image classification task. As can be seen, the experimental results presented in Fig. A1 are consistent with the findings in Sec.4 of the main paper, further reinforcing the validity of our conclusions. Specifically, as shown in Fig. A1-(a) and (b), model utility, measured by RA (retain accuracy) and TA (testing accuracy), decreases when external perturbations are applied to the evaluation sets compared to its original performance without perturbations. Meanwhile, Fig. A1-(c) and (d) show that unlearning effectiveness measured by UA (unlearning accuracy) and MIA-Efficacy, remains stable despite the presence of these perturbations on the forget set.

# C  PARAMETER SENSITIVITY ANALYSIS: PREDICTION MARGIN $\tau$ IN FORGET VECTOR LOSS

In **Fig. A2**, we provide the sensitivity analysis of $\tau$ on two datasets regarding two forgetting scenarios: class-wise forgetting and random data forgetting, where we varied $\tau$ from 0.0 to 2.2 with a step of

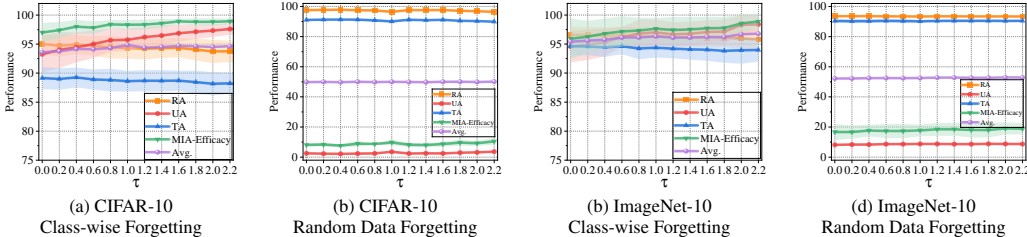

| (a) CIFAR-10 | (b) CIFAR-10 | (b) ImageNet-10 | (d) ImageNet-10 |
| Class-wise Forgetting | Random Data Forgetting | Class-wise Forgetting | Random Data Forgetting |

Figure A2: Sensitivity analysis of the nonnegative margin parameter $\tau$ for image classification under two unlearning scenarios on CIFAR-10 and ImageNet-10 using ResNet-18 and VGG-16, respectively. The unlearning performance metrics are reported as (a) TA (testing accuracy, blue curve), (b) RA (retain accuracy, orange curve), (c) UA (unlearning accuracy, red curve), and (d) MIA-Efficacy (green curve), (e) the average (Avg.) performance across TA, RA, UA, and MIA-Efficacy (purple curve). For class-wise forgetting scenario, the performance is averaged over 10 independent trials, with each trial focusing on forgetting one specific class from the dataset. Similarly, for random-data forgetting scenario, the performance is reported across 10 independent trials, where each trial targets the forgetting of a random subset of the dataset. The shaded regions represent the variance in performance across trials.

0.2. As can be seen, the forget objective is robust to variations in the nonnegative margin parameter $\tau$. When $\tau$ is set to 1, the performance across four metrics achieves an optimal tradeoff. Therefore, we choose $\tau = 1$ for our experiments.

# D  PERFORMANCE ON IMAGENET-10 USING VGG-16

Table A1: Performance overview of various MU methods for image classification under two unlearning scenarios on ImageNet-10 using VGG-16. Since Class-F (Kodge et al., 2024) is specifically designed for class-wise forgetting, its results do not apply to random data forgetting scenarios (n/a). Other results are reported in the format $a_{\pm b}$, where $a$ is the mean and $b$ denotes standard deviation $b$ over 10 independent trials. The performance gap against Retrain is indicated in (•), where a lower value is better. ↑ (or ↓) indicates that a higher (or lower) value is better. The best performance for each metric is highlighted in green, while the second-best performance is highlighted in red.

| MU Method | UA↑ | MIA-Efficacy↑ | RA↑ | TA↑ | Avg.Gap↓ | UA↑ | MIA-Efficacy↑ | RA↑ | TA↑ | Avg.Gap↓ |
|---|---|---|---|---|---|---|---|---|---|---|
| | Class-wise Forgetting, ImageNet-10, VGG-16 | | | | | Random Data Forgetting, ImageNet-10, VGG-16 | | | | |
| Retrain | $100.00_{\pm 0.00}(0.00)$ | $100.00_{\pm 0.00}(0.00)$ | $99.66_{\pm 0.16}(0.00)$ | $97.11_{\pm 0.82}(0.00)$ | 0.00 | $4.05_{\pm 0.45}(0.00)$ | $6.60_{\pm 1.07}(0.00)$ | $99.48_{\pm 0.07}(0.00)$ | $96.33_{\pm 0.38}(0.00)$ | 0.00 |
| FT | $39.66_{\pm 4.73}(60.34)$ | $55.76_{\pm 7.26}(44.24)$ | $99.78_{\pm 0.03}(0.13)$ | $97.27_{\pm 0.35}(2.35)$ | 26.77 | $1.35_{\pm 0.32}(2.70)$ | $4.67_{\pm 1.61}(1.93)$ | $99.36_{\pm 0.28}(0.12)$ | $96.54_{\pm 0.50}(0.21)$ | 1.24 |
| RL | $76.58_{\pm 11.64}(23.42)$ | $46.04_{\pm 33.71}(53.96)$ | $99.28_{\pm 0.20}(0.63)$ | $96.91_{\pm 0.55}(1.99)$ | 20.00 | $2.96_{\pm 0.42}(1.09)$ | $12.85_{\pm 4.25}(6.25)$ | $99.19_{\pm 0.16}(0.29)$ | $95.50_{\pm 0.90}(0.83)$ | 2.12 |
| GA | $46.61_{\pm 6.11}(53.39)$ | $49.15_{\pm 9.36}(50.85)$ | $99.35_{\pm 0.11}(0.56)$ | $95.60_{\pm 0.22}(0.68)$ | 26.37 | $0.18_{\pm 0.04}(3.87)$ | $2.97_{\pm 1.51}(3.63)$ | $99.86_{\pm 0.01}(0.38)$ | $97.47_{\pm 0.09}(1.14)$ | 2.23 |
| NegGrad+ | $49.56_{\pm 34.87}(50.44)$ | $64.27_{\pm 27.33}(35.73)$ | $99.08_{\pm 1.18}(0.58)$ | $96.47_{\pm 1.25}(0.64)$ | 21.84 | $0.60_{\pm 0.44}(3.45)$ | $3.90_{\pm 2.22}(2.70)$ | $99.68_{\pm 0.28}(0.20)$ | $97.12_{\pm 0.45}(0.79)$ | 1.79 |
| SalUn | $95.13_{\pm 1.79}(4.87)$ | $97.24_{\pm 0.17}(2.70)$ | $96.33_{\pm 0.25}(3.33)$ | $96.18_{\pm 1.10}(0.93)$ | 2.97 | $1.15_{\pm 0.40}(2.90)$ | $3.56_{\pm 1.12}(3.04)$ | $98.97_{\pm 1.56}(0.51)$ | $95.42_{\pm 2.10}(0.91)$ | 1.84 |
| SCRUB | $99.12_{\pm 0.14}(0.88)$ | $98.01_{\pm 0.51}(1.99)$ | $99.75_{\pm 0.03}(0.09)$ | $97.24_{\pm 0.16}(0.13)$ | 0.77 | $0.18_{\pm 0.08}(3.87)$ | $2.80_{\pm 1.33}(3.80)$ | $99.88_{\pm 0.03}(0.40)$ | $97.36_{\pm 0.15}(1.03)$ | 2.28 |
| Class-F | $71.19_{\pm 3.50}(28.81)$ | $63.57_{\pm 4.07}(36.43)$ | $61.55_{\pm 4.25}(38.11)$ | $59.40_{\pm 4.59}(37.71)$ | 35.27 | n/a | n/a | n/a | n/a | n/a |
| SSD | $91.50_{\pm 0.96}(8.50)$ | $91.30_{\pm 8.70}(8.70)$ | $99.48_{\pm 0.18}(0.18)$ | $97.00_{\pm 0.11}(0.11)$ | 4.38 | $0.88_{\pm 0.04}(3.17)$ | $5.35_{\pm 0.45}(1.25)$ | $99.36_{\pm 0.15}(0.12)$ | $96.40_{\pm 0.40}(0.07)$ | 1.15 |
| Ours | $87.23_{\pm 6.55}(12.77)$ | $91.41_{\pm 5.90}(8.59)$ | $94.77_{\pm 1.16}(5.14)$ | $94.04_{\pm 1.29}(0.88)$ | 6.85 | $2.27_{\pm 0.50}(1.78)$ | $6.13_{\pm 1.40}(0.47)$ | $98.29_{\pm 0.32}(1.19)$ | $95.82_{\pm 0.48}(0.51)$ | 0.99 |

In Tab. A1, we provide additional performance overview of different MU methods under two unlearning scenarios on ImageNet-10 using VGG-16. Considering the gain in unlearning effectiveness, the Avg. Gap with Retrain shows that the forget vector remains competitive, except for class-wise forgetting on (ImageNet-10, VGG-16). In this case, SCRUB achieves the best performance, as it leverages model distillation (Pérez-Cruz, 2008), a technique well-suited for class-wise forgetting.

# E  COMPONENT ANALYSIS: $\lambda_1$ AND $\lambda_2$

To verify the impact of each key component in the optimization problem (4) of the main paper, we analyze the nonnegative trade-off parameters $\lambda_1$ and $\lambda_2$ in Fig. A3. As can be seen in Fig. A3-(a) and (c), setting the retain loss regularization parameter $\lambda_1$ to 3 for CIFAR-10 using ResNet-18 and 5 for ImageNet-10 using VGG-16 in class-wise forgetting, along with the $\ell_2$-norm regularization parameter $\lambda_2 = 1$, enables our proposed method to achieve the highest average performance. Meanwhile, for the random data forgetting scenario, setting both $\lambda_1$ and $\lambda_2$ to 1 yields the best average performance for CIFAR-10 using ResNet-18 and ImageNet-10 using VGG-16.

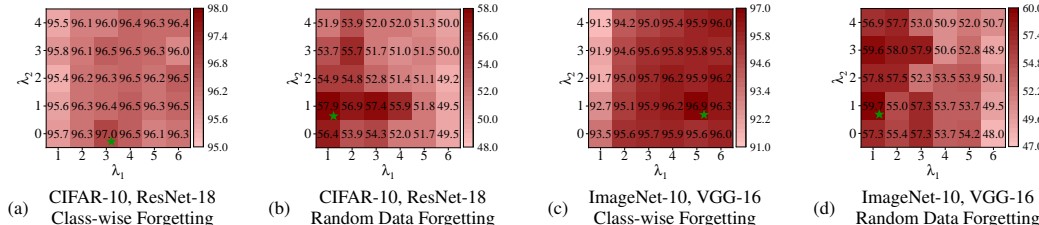

(a) CIFAR-10, ResNet-18 Class-wise Forgetting

(b) CIFAR-10, ResNet-18 Random Data Forgetting

(c) ImageNet-10, VGG-16 Class-wise Forgetting

(d) ImageNet-10, VGG-16 Random Data Forgetting

Figure A3: Sensitivity analysis of the nonnegative hyper-parameters $\lambda_1$ (ranging from 1 to 5 with an interval of 1) and $\lambda_2$ (ranging from 0 to 4 with an interval of 1) for image classification under two unlearning scenarios on CIFAR-10 and ImageNet-10 using ResNet-18 and VGG-16, respectively. The unlearning performance is reported using *the average (avg.) performance across UA, RA, TA, and MIA-Efficacy*, with a green star (★) marking the chosen parameter scheme ($\lambda_1$ and $\lambda_2$) in our experiments. The color bar on the right represents a gradient scale from light to dark red, indicating the range of values (0 to 100%) in the heatmap. The integer within each cell represents the performance (%) given a combination of $\lambda_1$ and $\lambda_2$.

Table A2: The efficiency profile of different MU methods across two metrics for image classification under two unlearning scenarios on CIFAR-10 (RestNet-18) and ImageNet-10 (VGG-16 and ViT-Base). Results are reported in terms of run-time efficiency (RTE) measured in *minutes* for the overall training phase and parameter efficiency denoted by parameter number (Param.#) in Million (M), where a smaller value is favored for each metric.

| MU Method | Class-wise Forgetting | | | | | | Random Data Forgetting | | | | | |
|---|---|---|---|---|---|---|---|---|---|---|---|---|
| | CIFAR-10, ResNet-18 | | ImageNet-10, VGG-16 | | ImageNet-10, ViT-Base | | CIFAR-10, ResNet-18 | | ImageNet-10, VGG-16 | | ImageNet-10, ViT-Base | |
| | RTE | Param. # | RTE | Param. # | RTE | Param. # | RTE | Param. # | RTE | Param. # | RTE | Param. # |
| Retrain | 81.30 | 11.17 | 118.89 | 15.31 | 301.98 | 85.81 | 103.71 | 11.17 | 116.10 | 15.31 | 290.60 | 85.81 |
| FT | 4.56 | 11.17 | 2.99 | 15.31 | 15.40 | 85.81 | 2.28 | 11.17 | 2.95 | 15.31 | 14.47 | 85.81 |
| RL | 5.61 | 11.17 | 3.58 | 15.31 | 33.00 | 85.81 | 4.75 | 11.17 | 3.59 | 15.31 | 32.84 | 85.81 |
| GA | 1.16 | 11.17 | 0.30 | 15.31 | 2.10 | 85.81 | 1.15 | 11.17 | 0.31 | 15.31 | 3.10 | 85.81 |
| NegGrad+ | 4.72 | 11.17 | 3.21 | 15.31 | 16.70 | 85.81 | 4.22 | 11.17 | 3.17 | 15.31 | 16.40 | 85.81 |
| SalUn | 2.81 | 5.59 | 2.56 | 7.66 | 18.73 | 42.91 | 3.01 | 5.59 | 2.85 | 7.66 | 18.19 | 42.91 |
| SCRUB | 1.23 | 11.17 | 1.62 | 15.31 | 33.45 | 85.81 | 1.53 | 11.17 | 1.55 | 15.31 | 22.27 | 85.81 |
| Class-F | 1.21 | 11.17 | 2.10 | 15.31 | 2.86 | 85.81 | n/a | n/a | n/a | n/a | n/a | n/a |
| SSD | 1.20 | 11.17 | 2.10 | 15.31 | 2.21 | 85.81 | 1.15 | 11.17 | 2.00 | 15.31 | 2.22 | 85.81 |
| Ours | 2.37 | 0.03 | 6.15 | 0.15 | 4.80 | 0.15 | 6.15 | 0.03 | 6.50 | 0.15 | 8.58 | 0.15 |

# F    EFFICIENCY ANALYSIS

To demonstrate the efficiency profile of various MU methods under different metrics, we compare the runtime costs of forget vector calculation with those of other model-based unlearning methods, and the total number of updated parameters in the training phase. Specifically, following Fan et al. (2023), we use run-time efficiency (RTE) as an evaluation metric, which measures the computation time of applying an MU method in minutes. To ensure a fair comparison, we report the runtime of each approximate MU method within 10 epochs since the number of iterations varies across different MU methods. Additionally, inspired by Zhang et al. (2024b), we compare the number of trainable parameter number (Param.#). Notably, all evaluations are performed in the same computational environment with an NVIDIA A6000 GPU, ensuring fair and reliable comparisons by maintaining a consistent batch size across all MU methods. The corresponding results can be found in Tab. A2, and we can draw the following observation: 1) By comparing the training time of model-based methods with our proposed approach using RTE within the same iterations, we observe that the forget vector method achieves competitive RTE compared to some efficient approximate model-based unlearning baselines and remains faster than retraining. Besides, it is worth noting that the forget vector method optimizes fewer parameters, owing to its input-level design and perturbation space is significantly lower-dimensional. This trade-off underscores the efficiency of our forget vector in handling parameter scaling. For instance, when comparing forgetting cases on ImageNet-10 using VGG-16 and ViT-Base models, we find that as the parameter count increases from "15.31M" in VGG-16 to "85.81M" in ViT-Base, the optimization time for most baseline methods increases significantly. In contrast, our method maintains a consistent efficiency level or even achieves a shorter runtime, highlighting a key advantage of our forget vector approach. 2) From a memory-efficient perspective in real-world application, existing model-based MU methods often necessitate a series of operations (*e.g.*, fine-tuning) on the already-trained model for each forgetting request. This necessitates storing a separate model version for every request, leading to substantial storage overhead. In contrast, our approach optimizes an input-level universal perturbation, where only the "forget vector" associated with the data to be forgotten is stored with original model remains intact. Since each forget vector has the same dimensionality as the input image (*e.g.*, 0.03M for CIFAR-10 and 0.15M for ImageNet) and

is significantly smaller than the full model, our approach offers a considerable advantage in storage efficiency for practical applications.

## G  ADDITIONAL VISUALIZATION THROUGH AN INPUT SALIENCY LENS

In Figs. A4 and A5, we provide additional visualization results through input saliency map for different methods against forget images and retain images, respectively, using Grad-CAM (Gradient-weighted Class Activation Mapping) (Bengio et al., 2013). Consistent with the main paper, we highlight the salient pixels (*i.e.*, regions most influential to model predictions) under four scenarios: (1) the original model (without unlearning), (2) Retrain, (3) SCRUB-based unlearning, and (4) forget vector-based unlearning. For the first three scenarios, saliency maps are generated on raw forget/retain images, whereas for the last scenario, they are applied to images perturbed by the forget vector.

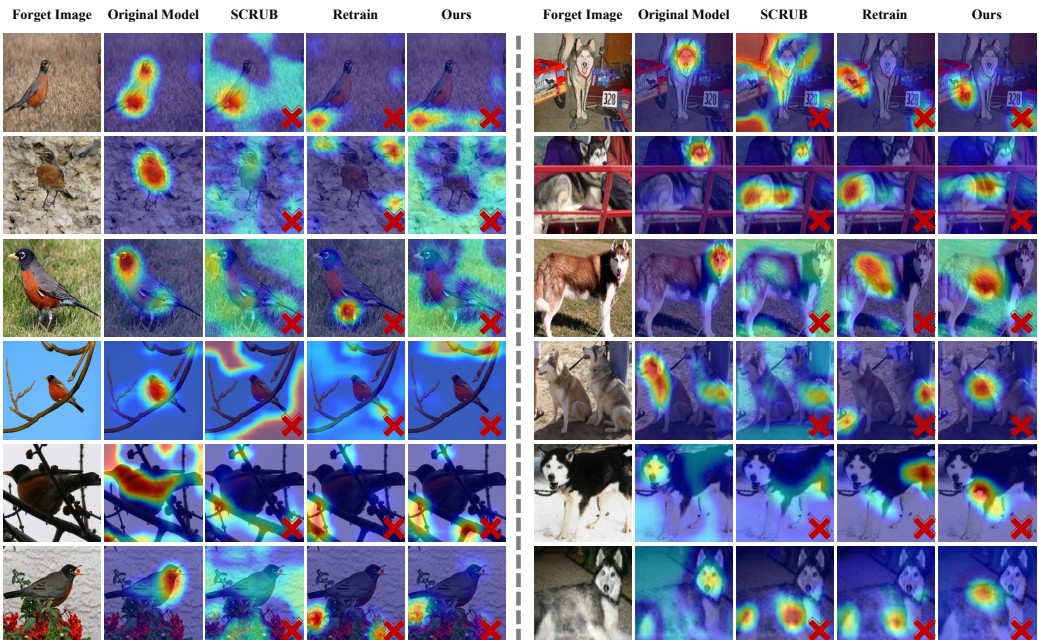

Figure A4: Gradient-based saliency map visualized via Grad-CAM for different MU methods against **forget images**. The highlighted areas (marked in red) indicate regions most influential to model prediction, and the red cross mark (✖) indicates that corresponding methods effectively unlearn the input forget images.

## H  BROADER IMPACTS AND LIMITATIONS

**Broader impacts.**  Our study on the forget vector introduces a novel, data-driven machine unlearning approach that offers significant potential across various domains. ① Privacy Preservation and Regulatory Compliance: With increasing global regulations like GDPR mandating the right to be forgotten, our method enables effective unlearning of specific data points without requiring full model retraining, which is especially valuable for industries handling sensitive data, such as healthcare, finance, and personalized recommendation systems. By enabling compliant data removal with minimal computational overhead, our approach strengthens data privacy practices while maintaining model integrity. ② Adaptability Across Tasks: The compositional nature of forget vectors allows for the unlearning of previously unseen tasks by combining class-specific forget vectors. This adaptability extends the method's applicability, where our method is able to respond more quickly when faced with new random unlearning tasks. ③ Efficiency in Unlearning and Model Storage: Unlike model-based MU approaches that require significant computational resources and model storage for each unlearning request, our method provides an energy-efficient alternative with less trainable parameters and low storage of optimized forget vector. This efficiency is particularly beneficial for organizations deploying large-scale AI models, where frequent updates or compliance-driven data deletions could otherwise be prohibitively expensive, contributing to the sustainability of AI development.

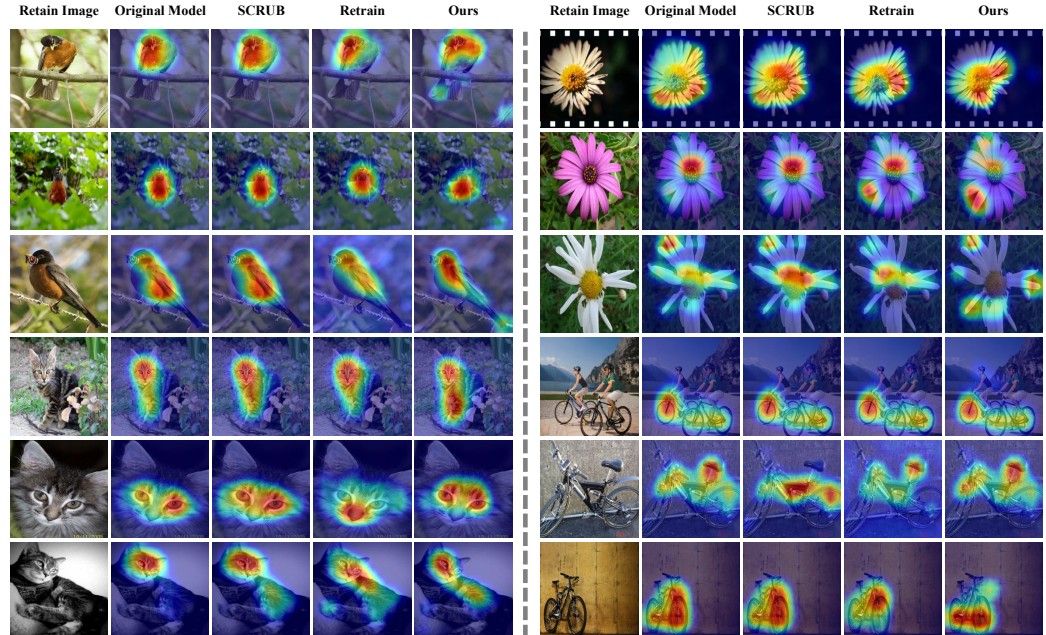

Figure A5: Gradient-based saliency map visualization using Grad-CAM for different MU methods against **retain images**.

**Limitations.**  While our forget vector approach presents compelling advantages, it also comes with certain limitations that need to be addressed for broader adoption: ① Vulnerability to Adversarial Attacks: Since our method relies on input perturbations rather than direct model modifications, it may be vulnerable to white-box adversarial attacks where an attacker has access to the model and understands the forget vector mechanism. These adversaries could potentially reconstruct forgotten data or design countermeasures to bypass unlearning. Future work should focus on strengthening robustness against such attacks through adversarially hardened perturbation strategies. ② Computational Cost of Compositional Unlearning: The generation of new forget vectors through compositional operations requires access to pre-trained class-wise forget vectors. In scenarios where these vectors are unavailable, applying our method at scale could introduce overhead. ③ Lack of Theoretical Guarantees: While empirical results demonstrate the effectiveness of our method, formal theoretical guarantees on its unlearning performance and robustness remain an open challenge. Future work should focus on establishing rigorous mathematical foundations for the forget vector framework. ④ Generalization to Other Domains: Our current study focuses on image classification, and its applicability to other domains, such as natural language processing or image generation, remains to be explored. Investigating the feasibility of forget vectors in diverse machine learning tasks is an important direction for future research.

# I  LLM USAGE

In preparing this manuscript, large language models (LLMs) were employed exclusively for language-related refinement, such as grammatical correction, stylistic polishing, and improvement of readability. The LLMs did not contribute to the conceptualization, methodological design, experimental implementation, data analysis, or interpretation of results.

