# OpenReview forum: "Forget Vectors at Play: Universal Input Perturbations Driving Machine Unlearning in Image Classification"
_ICLR.cc/2026/Conference — Submitted to ICLR 2026_

### Official Review · Reviewer_xw9H · 2025-10-24

**Soundness:** 2
**Presentation:** 3
**Contribution:** 3
**Rating:** 4
**Confidence:** 3

**Summary:**

This paper reframes machine unlearning for image classification as a data-based operation. Instead of updating weights, the authors learn a universal, input-agnostic perturbation, “forget vector”, that is added to inputs to degrade predictions on the forget set while preserving utility on retain and test data. They formulate an optimization with an untargeted margin loss on forget set and a cross-entropy retain regularizer on retain set, plus an L2 norm penalty. The model itself stays frozen. They further propose compositional unlearning by linearly combining class-wise forget vectors to address random data deletion. Experiments on CIFAR-10 and ImageNet-10 with ResNet-18 / VGG-16 / ViT-Base compare against Retrain and several approximate MU baselines, reporting competitive unlearning performance.

**Strengths:**

1. The paper is well-written and easy to follow.

2. Achieving unlearning by keeping the model fixed and using a visual-prompting-like input perturbation is novel, offering a fresh perspective on approximate machine unlearning.

3. The idea of compositional Forget Vector arithmetic is interesting.

4. Grad-CAM visualizations provide intuitive evidence supporting the reported effects.

**Weaknesses:**

1. The proposed method that perturbs only the input via a Forget Vector by a linear operation in image input space has inherent limitations.

    * Effectiveness: Although the paper achieves competitive unlearning accuracy, this comes at the cost of degraded RA and TA due to the added perturbation.

    * Efficiency: According to Table A2 (RTE), the Forget Vector is not clearly superior to other model-based unlearning methods in runtime, and its performance does not significantly surpass them.

    Therefore, for a model maintainer, there may be insufficient reason to prefer the Forget Vector approach as the unlearning mechanism. Exploring hybrid approaches that combine Forget Vectors with other unlearning methods could better demonstrate its value.

2. The unlearning settings (only single-class forgetting and 10% random sample forgetting) considered are limited. A robust unlearning method should handle arbitrary numbers of classes and larger proportions of random samples. This may be particularly challenging for universal perturbations, as in the Forget Vector method.

3. The application scope in this paper is narrow, focusing on image classification. Unlearning is also relevant in image generation models, large language models, and multimodal models. The transferability of this approach remains unknown. For example, in vision–language models, whether an image-side Forget Vector can achieve comparable forgetting effects is an important unresolved question.

**Questions:**

1. How does Forget Vector perform in the following scenarios aimed at addressing the aforementioned weaknesses: (i) its integration with other unlearning methods, (ii) evaluation under more challenging unlearning settings, and (iii) evaluation on vision–language models?

2. In the experiments related to Figure 3, the combination coefficient $w_1=-1$ for Forget Vectors is negative. While this is plausible from an optimization perspective, it seems at odds with the intended semantics of a forget vector. Could you provide a clear explanation for this phenomenon?

3. In Table 1, the green/red highlighting appears to be based on performance gaps, which might conflict with the up/down arrows (e.g., TA in the Random Data Forgetting, ImageNet-10, ViT-Base setting). It would help to clearly state the convention in the caption to avoid ambiguity.

---

> ### Author Response · Authors · 2025-11-21
> **To Reviewer xw9H**
>
> # 1. Response to the Concern on Method Effectiveness and Efficiency.
> We appreciate the reviewer’s thoughtful assessment. Below we clarify two key points:
>
> (1) On the effectiveness and utility degradation:
>  While the Forget Vector (FV) introduces a perturbation in input space, the perturbation is explicitly optimized to be minimal and utility-preserving (Equation 4). As shown in multiple analyses (Tables 1–2 and cosine similarity in **Table R1** below), Forget Vector preserves high representation similarity for retain data, and their GradCAM saliency maps remain nearly unchanged (Figures. 4, A5). This explains why RA/TA decreases only slightly, an inherent trade-off for input-level unlearning that avoids modifying model parameters. Importantly, FV still achieves competitive overall Average Gap, comparable to model-based baselines, confirming that the utility degradation is limited and well-controlled.
> | Set| Cosine Similarity   |
> |------------------|-------------------|
> | Retain Set      |0.88      |
> | Forget Set     | 0.21         |
>
> **Table R1**: Cosine similarity between original and perturbed inputs across different data partitions, showing that the forget vector preserves representations for retain samples while inducing strong shifts only for the forget set.
>
> (2) On efficiency relative to model-based methods:
> Model-based unlearning methods must update millions of model parameters for every forgetting request (e.g., 15M–85M for VGG/ViT) and often need to store separate model versions for different unlearning tasks. In contrast, FV optimizes only a single vector with the dimensionality of the input (0.03M for CIFAR-10; 0.15M for ImageNet-10) while keeping the model weights completely unchanged. This not only eliminates the need to modify or store multiple models, but also makes the optimization space significantly smaller.
>
> Furthermore, the computation cost of FV is dominated by optimizing the forget vector—not by updating model parameters. This becomes a larger advantage as model size increases, because the overhead of model-based methods grows with the parameter count, whereas FV’s cost remains tied only to the input dimension. When extended to compositional unlearning, the cost becomes even smaller, since only a few scalar coefficients need to be optimized. As a result, FV offers dramatically lower storage and computational cost, making it far more scalable under repeated or dynamic unlearning requests—scenarios where model-based MU methods face large cumulative overhead.
>
> Moreover, we include additional efficiency analysis for Compositional Unlearning via Class-wise Forget Vectors (CU-FV) in **Table R2** below. In addition, we further evaluate our approach on a larger backbone (ViT-Large) under the random-data forgetting scenario with a 10% forget ratio on ImageNet-10. As shown in **Table R2**, CU-FV achieves up to 19× faster runtime than FV while maintaining competitive unlearning and utility performance.
>
> Importantly, **Table R2** also highlights a key trend: as the backbone model becomes larger, the efficiency advantage of our data-based methods (FV and CU-FV) becomes increasingly significant. Model-based MU methods must update tens or hundreds of millions of parameters (e.g., 151M-303M for ViT-Large), causing their runtime to grow substantially with model size.
>
> | MU Method | UA↑ | MIA-Efficacy↑ | RA↑ | TA↑ | Avg.Gap↓ | RTE | Param.# |
> |-----------|------|----------------|------|-------|-----------|------|----------|
> | Retrain   | 0.69 | 93.53         | 99.27 | 99.00 | 0.00      | 18.50 | 303.40(M) |
> | FT        | 0.61 | 97.80         | 97.33 | 98.00 | 1.44      | 9.30  | 303.40(M) |
> | Salun     | 0.93 | 98.80         | 99.00 | 98.04 | 1.67      | 7.59  | 151.70(M) |
> | FV        | 0.75 | 98.30         | 98.15 | 99.15 | 1.51      | 5.71  | 0.15(M)   |
> | CU-FV     | 0.85 | 98.50         | 99.11 | 99.80 | 1.50      | 0.30  | 10        |
>
>
> **Table R2**: Performance and efficiency overview of various MU methods under random data forgetting scenario on ImageNet-10 using ViT-Large, across the metrics: UA, RA, TA, MIA-Efficacy, Avg. Gap vs. Retrain, run-time efficiency (RTE) and parameter number (Param.#), where RTE is measured in minutes and M refers to Million.

---

> > ### Author Response · Authors · 2025-11-21
> > **To Reviewer xw9H**
> >
> > # 2. Response to the Concern on Practical Value and Hybrid Approaches.
> >
> > We thank the reviewer for the thoughtful suggestion. Our goal is to address a complementary and increasingly important unlearning setting, where model parameters do not need to update. In these scenarios, Forget Vectors provide a capability that model-based methods fundamentally cannot offer. In addition, Compositional Unlearning via FVs provide clear advantages in runtime and parameter efficiency, especially for large-scale models, since only a single perturbation vector is optimized while the model remains untouched. This makes FV preferable for repeated or dynamic unlearning requests, where maintaining multiple fine-tuned model versions is impractical.
> >
> > We agree that hybrid approaches combining FV with model-based MU are a promising direction. Although FV does not modify model parameters, it can naturally complement model-based MU. FV provides an explicit input-space forgetting direction that can be used to guide or accelerate parameter-update MU methods, or serve as a lightweight post-hoc safety layer to suppress residual memorization. We agree that such hybrid designs are promising, and will add this discussion as a direction for future work.

---

> > > ### Author Response · Authors · 2025-11-21
> > > **To Reviewer xw9H**
> > >
> > > # 3. Response to the concern on the limited unlearning settings (single-class & 10% random forgetting).
> > > We thank the reviewer for raising this important point. We agree that a robust unlearning mechanism should scale beyond single-class forgetting and the 10% random-forgetting setting. To address this, we have conducted additional experiments to evaluate Forget Vectors (FV) under more challenging and diverse unlearning configurations, and summarize our findings below.
> > >
> > > To evaluate how our approach scales with the number of classes in the forget set, we conduct experiments on ImageNet-10 using the ViT-Base backbone. Specifically, we measure performance as the model is tasked to forget 1, 2, and 3 classes simultaneously under the class-wise forgetting setting. The results can be found in **Table R3**.
> > >
> > > ### → 1 class
> > >
> > > | MU Method | UA ↑   | MIA-Efficacy ↑ | RA ↑   | TA ↑   | Avg. Gap ↓ |
> > > |----------|--------|----------------|--------|--------|------------|
> > > | Retrain  | 100.00 | 100.00         | 99.97  | 99.85  | 0.00       |
> > > | FT       | 42.79  | 40.78          | 99.96  | 99.61  | 29.17      |
> > > | SalUn    | 93.27  | 94.00          | 98.22  | 98.00  | 4.08       |
> > > | Ours     | 95.92  | 99.40          | 99.13  | 99.26  | 1.53       |
> > >
> > > ---
> > >
> > > ### → 2 classes
> > >
> > > | MU Method | UA ↑   | MIA-Efficacy ↑ | RA ↑   | TA ↑   | Avg. Gap ↓ |
> > > |----------|--------|----------------|--------|--------|------------|
> > > | Retrain  | 100.00 | 100.00         | 99.95  | 99.80  | 0.00       |
> > > | FT       | 48.62  | 46.45          | 99.91  | 99.48  | 26.32      |
> > > | SalUn    | 92.80  | 93.40          | 97.15  | 98.00  | 4.60       |
> > > | Ours     | 95.67  | 97.35          | 98.45  | 98.56  | 2.43       |
> > >
> > > ---
> > >
> > > ### → 3 classes
> > >
> > > | MU Method | UA ↑   | MIA-Efficacy ↑ | RA ↑   | TA ↑   | Avg. Gap ↓ |
> > > |----------|--------|----------------|--------|--------|------------|
> > > | Retrain  | 100.00 | 100.00         | 99.90  | 99.75  | 0.00       |
> > > | FT       | 47.13  | 45.90          | 99.90  | 99.50  | 27.01      |
> > > | SalUn    | 93.50  | 94.20          | 97.15  | 97.50  | 4.32       |
> > > | Ours     | 95.94  | 93.20          | 97.21  | 96.33  | 4.24       |
> > >
> > > **Table R3**. Performance overview of various MU methods for image classification in class-wise forgetting scenario on ImageNet-10 using ViT-Base with different number of forgetting classes.
> > >
> > >
> > > **On scalability across different unlearning ratios.**
> > > We further evaluated our method under different forget ratios (10%, 20%, 30%) by performing random data forgetting on ImageNet-10 with a ViT-Base backbone. As shown in **Table R4**, our approach consistently suppresses forgotten data while preserving high accuracy on retained data, demonstrating robustness and scalability across varying unlearning demands.
> > >
> > > ### → 10%
> > > | Forgetting Ratio | MU Method | UA ↑ | MIA-Efficacy ↑ | RA ↑ | TA ↑ | Avg. Gap ↓ |
> > > |------------------|-----------|------|----------------|------|------|------------|
> > > | 10% | Retrain | 1.41 | 93.57 | 99.07 | 99.27 | 0.00 |
> > > | 10% | FT      | 1.38 | 96.40 | 99.60 | 99.10 | 0.89 |
> > > | 10% | Salun   | 0.67 | 95.80 | 99.65 | 98.27 | 1.14 |
> > > | 10% | Ours    | 1.08 | 91.40 | 98.97 | 99.10 | 0.69 |
> > > ---
> > > ### → 20%
> > > | Forgetting Ratio | MU Method | UA ↑ | MIA-Efficacy ↑ | RA ↑ | TA ↑ | Avg. Gap ↓ |
> > > |------------------|-----------|------|----------------|------|------|------------|
> > > | 20% | Retrain | 1.65 | 96.03 | 98.35 | 98.00 | 0.00 |
> > > | 20% | FT      | 1.93 | 94.00 | 99.26 | 99.00 | 1.05 |
> > > | 20% | Salun   | 0.88 | 94.10 | 99.05 | 98.50 | 0.98 |
> > > | 20% | Ours    | 1.80 | 95.50 | 98.11 | 98.20 | 0.28 |
> > > ---
> > > ### → 30%
> > > | Forgetting Ratio | MU Method | UA ↑ | MIA-Efficacy ↑ | RA ↑ | TA ↑ | Avg. Gap ↓ |
> > > |------------------|-----------|------|----------------|------|------|------------|
> > > | 30% | Retrain | 1.95 | 96.71 | 99.04 | 99.00 | 0.00 |
> > > | 30% | FT      | 1.77 | 95.80 | 98.84 | 98.20 | 0.52 |
> > > | 30% | Salun   | 0.90 | 96.10 | 98.95 | 98.30 | 0.61 |
> > > | 30% | Ours    | 2.38 | 97.00 | 98.33 | 98.20 | 0.56 |
> > >
> > >
> > > **Table R4**. Scalability of various unlearning methods across varying forgetting ratios on ImageNet10 using ViT-Base.

---

> > > > ### Author Response · Authors · 2025-11-21
> > > > **To Reviewer xw9H**
> > > >
> > > > # 4. Response to the concern about narrow application scope.
> > > > We appreciate the reviewer’s point regarding the broader applicability of machine unlearning beyond image classification. Our current work focuses on supervised image classifiers because this setting provides a clean and controlled environment to isolate and study the core behavior of input-level, model-agnostic unlearning—which is precisely what Forget Vectors (FVs) aim to achieve.
> > > >
> > > > That said, we fully agree that extending FV to image generation models, multimodal models, and large language models is an important direction. Conceptually, Forget Vectors belong to the family of input-space reprogramming / universal perturbation techniques, which have already shown transferability to multiple domains (e.g., diffusion models, LLM prompting, and multimodal prompting). This suggests that FV may also generalize beyond image classification, although concrete demonstrations remain to be established.
> > > >
> > > > For example, in a vision–language model (e.g., CLIP or BLIP), one could imagine learning:
> > > > * image-side forget vectors to erase visual patterns tied to sensitive concepts, or
> > > > * paired image–text forget perturbations to jointly de-emphasize cross-modal associations.
> > > >
> > > > However, because these models involve richer interactions (e.g., alignment between visual embeddings and textual embeddings), additional mechanisms—such as cross-modal consistency constraints—would likely be needed to ensure effective forgetting without harming alignment.
> > > > Given these complexities, we view cross-domain generalization as an exciting but nontrivial challenge. We will add a dedicated discussion in the paper and explicitly highlight this as a major avenue for future research.
> > > >
> > > >
> > > > # 5.Response to the integration with other MU methods, robustness under harder settings, and VLM evaluation.
> > > > We appreciate the reviewer’s insightful questions. Below we provide clarifications and new results addressing all three aspects.
> > > >
> > > > **(i)  On integrating Forget Vectors (FV) with model-based unlearning methods.**
> > > >
> > > > FV is inherently model-agnostic and operates in the input space, while most existing MU methods (e.g., fine-tuning-based FT/RL/GA, saliency-based SalUn/SSD, and distillation-based SCRUB) operate in the weight space. Because these two paradigms manipulate orthogonal components of the system, FV can be naturally combined with model-based MU.
> > > > There are two practical hybrid designs:
> > > >
> > > > (a) FV → Model-based MU
> > > >  Use FV to perform fast, data-level unlearning as a first step, then apply a lightweight model-based MU method to refine the model.
> > > >  * FV quickly handles large forgetting requests.
> > > > * Model-based MU corrects fine-grained residual memorization.
> > > >
> > > > (b) Model-based MU → FV
> > > >  First apply a model-based method to reduce memorization, then learn a small FV to handle remaining data points, serving as a plug-in residual unlearner.
> > > >
> > > > Since FV does not modify model weights, such hybrid schemes do not conflict with existing MU methods. We highlight this hybrid direction as promising future work.
> > > >
> > > > **(ii) On evaluation under more challenging unlearning settings.**
> > > > We thank the reviewer for the suggestion. We have now included new experiments on:
> > > >
> > > > (a) Increasing number of forgotten classes (1 → 2 → 3 classes).
> > > >
> > > > Results in **Table R3** (ViT-Base, ImageNet-10) show that FV scales smoothly from 1 to 3 forgotten classes, and FV consistently outperforms FT and matches or exceeds SalUn.
> > > >
> > > > (b) Increasing random forgetting ratio (10% → 20% → 30%)
> > > > Results in **Table R4** demonstrate that FV maintains strong unlearning effectiveness with different forgetting ratio.
> > > > These results confirm that FV remains robust even when unlearning becomes more difficult.
> > > >
> > > > **(iii) On evaluation for vision–language models (VLMs)**
> > > > We agree that applying FV to VLMs is an important direction. Extending FV to VLMs requires rethinking the modality alignment pipeline, for example:
> > > > * Whether an image-side FV can induce forgetting in the joint embedding space;
> > > > * Whether separate FV components are needed for the vision tower and for the cross-modal fusion module;
> > > > * How perturbations on the image side propagate into the text–image alignment loss.
> > > >
> > > > These aspects go beyond the scope of this paper, as they require new architectures, loss formulations, and potentially task-specific constraints. Nevertheless, we have added a dedicated discussion in the paper and identified this as a key avenue for future exploration.

---

> > > > > ### Author Response · Authors · 2025-11-21
> > > > > **To Reviewer xw9H**
> > > > >
> > > > > # 6. Response to the question about negative combination coefficients in Figure 3.
> > > > > Thank you for raising this point. Although a “forget vector” is conceptually described as pushing the input toward a direction that induces forgetting, in practice the linear combination of class-wise forget vectors does not need to be restricted to positive weights. A negative coefficient simply means that the optimizer finds that moving slightly opposite to a class-specific forget direction helps reduce the overall gap between unlearning effectiveness (UA) and utility retention (RA).
> > > > >
> > > > > This happens because:
> > > > >
> > > > > (1) Each class-wise forget vector is not perfectly orthogonal.
> > > > >  Different classes share visual features, so their forget directions partially overlap. A small negative weight can “cancel out” excessive perturbation from another forget vector, producing a more balanced δ(w).
> > > > >
> > > > >
> > > > > (2) The goal is not semantic interpretation of each δₖ but minimizing the global unlearning objective.
> > > > >  The coefficients w1,w2​ are optimized purely to minimize the combined loss (forgetting + retaining). If the best trade-off requires reducing the influence of a particular δₖ, the optimizer naturally assigns a negative weight.
> > > > >
> > > > >
> > > > > (3) Negative coefficients do not imply “remembering.”
> > > > >  They simply adjust the final perturbation so that UA is high (effective forgetting), RA and TA remain stable (utility retention), and Avg. Gap relative to Retrain is minimized.
> > > > >
> > > > > # 7. Response to the conflict with the up/down arrows.
> > > > > Thank you for pointing out this potential ambiguity.
> > > > >
> > > > >  We clarify that two different pieces of information are presented in Table 1:
> > > > >
> > > > > (1) Raw metric values (e.g., TA, RA, UA, MIA-Efficacy), where up/down arrows indicates their intrinsic semantic direction. These arrows only describe how to interpret the metric itself.
> > > > >
> > > > > (2) Green/red highlighting, The highlighting does not refer to the raw metric values. It is applied solely to the performance gaps relative to Retrain, which serves as the gold standard for machine unlearning. This provides a uniform and fair ranking across metrics, regardless of whether higher or lower values of the metric are desirable.
> > > > >
> > > > > Thus, there is no conflict: arrows indicate each metric’s semantic direction, whereas green/red highlighting reflects closeness to the gold-standard Retrain based solely on Avg. Gap; we will clarify this convention in the caption.

---

### Official Review · Reviewer_oFB6 · 2025-10-27

**Soundness:** 3
**Presentation:** 3
**Contribution:** 2
**Rating:** 6
**Confidence:** 3

**Summary:**

This paper proposes a machine unlearning method that introduces an augmentation vector to perturb input data, preventing a trained machine learning model from making correct predictions on the designated forget datasets. The augmentation vector is treated as a set of trainable variables, optimized through a combined forget and retain loss functions to balance unlearning performance and knowledge retention. Experimental results demonstrate that the proposed approach is effective for both class-level and random unlearning tasks. However, the method also leads to noticeable performance degradation on the retain and test datasets.

**Strengths:**

1. The paper is clearly written and presents a well-motivated discussion on how the proposed method differs from existing unlearning approaches.
2. The experimental evaluation is comprehensive and includes standard benchmarks and metrics commonly used in the literature for assessing unlearning algorithms.

**Weaknesses:**

I have several concerns regarding the novelty of the proposed method and its impact on performance degradation over the retain datasets.

Although the paper claims the approach to be data-based, the method essentially trains an operator to modify the input, which is conceptually similar to techniques explored in the visual prompting literature, as acknowledged in the background section. The primary difference lies in the definition of the encoding function $f$, where previous works employ linear or nonlinear projections, while this method defines $f(x)=x+w$. This formulation represents a relatively minor variation rather than a fundamentally new idea.

The experimental results indicate a noticeable degradation in performance on the retain and test datasets compared with other methods. This suggests the limitation of augmenting inputs via simple vector addition. It would strengthen the paper to include an analysis of the geometric properties of the learned augmentation vectors relative to the training data, to better illustrate how they contribute to domain shift.

BTW: the green and red tags indicating the best and second-best performance in Table 1 appear to be mislabeled. E.g, RA on the random data forgetting task for ImageNet-10.

**Questions:**

Can the authors show what the vectors learned captured, is it a vector that move input representation to the representation area of retain data points (maybe a specific class)?

---

> ### Author Response · Authors · 2025-11-21
> **To Reviewer oFB6**
>
> # 1-1. Response to concerns regarding novelty and retain-set performance degradation
>  Our method introduces a new data-based unlearning paradigm that differs fundamentally from prior model-based MU approaches. Existing MU methods modify model parameters to enforce forgetting, whereas our approach is the first to show that a universal, input-agnostic perturbation can be optimized to achieve unlearning while keeping the model entirely fixed.  Our method intentionally avoids modifying model parameters, enabling post-hoc, on-demand unlearning in frozen or resource-constrained settings. While residual information in the weights is a valid concern, our approach focuses on behaviorally suppressing the model’s ability to recognize forgotten data through input-level perturbations, rather than physically altering the parameters.
> This design offers key practical benefits:
> * Reusability & utility preservation: The model remains unchanged, ensuring consistent performance on future downstream tasks.
> * Dynamic unlearning: Forget requests can be handled on-the-fly without retraining or versioning. Forgetting is reversible—simply remove the applied vector to restore recognition.
> * Lightweight storage: Only compact forget vectors are stored per request, avoiding multiple large model checkpoints. Therefore, we argue that while we do not remove forgotten data from model weights, our model-agnostic, input-level strategy is well-suited for practical deployment scenarios, especially those requiring flexibility, efficiency, and test-time control.
>
> # 1-2. Response to concerns regarding retain-set performance degradation.
> Regarding performance degradation on the retain set, a certain level of utility drop is expected because unlearning is performed through perturbing the inputs rather than modifying the model. However, as shown in Table 1, our method achieves top-tier unlearning effectiveness (UA and MIA-Efficacy) and maintains overall competitiveness in terms of Avg. Gap vs. Retrain, often ranking among the best two methods across settings. The optimization objective in Equation (4)  explicitly regularizes retain-set preservation, and the perturbation magnitude is constrained by the L2 penalty, ensuring that the performance drop remains controlled.
>
>
> # 2. Response to the concern on novelty and relation to visual prompting literature.
> Thank you for raising this point. Indeed, we explicitly acknowledge this connection in the paper. The core novelty of our paper does not lie in redefining the encoding function, but in recasting machine unlearning as an input‐space optimization problem, which has not been explored in prior work. Visual prompting aims to improve model performance on new tasks, whereas our forget vector is specifically optimized to erase targeted information while simultaneously preserving utility on retain data—an objective fundamentally different from prior prompting formulations.
>
> Moreover, while visual prompting is designed for task adaptation, our method targets an entirely different goal—data forgetting. The forget vector supports class-wise and random-data unlearning, generalizes to shifted forget sets, and enables compositional unlearning through forget vector arithmetic, allowing new unlearning tasks to be constructed without any retraining or re-optimization. These capabilities are specific to the unlearning setting and are not exhibited by prior prompting-based approaches.
>
> In summary, while we build on the intuition that input perturbations can guide a fixed model (similar to prompting), the core contribution lies in formulating machine unlearning as an input-based optimization problem, designing an MU-specific objective, and showing—for the first time—that unlearning can be achieved and composed entirely through universal input perturbations without updating model parameters.

---

> > ### Author Response · Authors · 2025-11-21
> > **To Reviewer oFB6**
> >
> > # 3. Response to the Concern on Utility Drop and Request for Geometric Analysis.
> > We appreciate the reviewer’s observation. Indeed, as a data-based MU method, our forget vector introduces an input-space perturbation, which naturally shifts certain samples slightly away from the training distribution, this can lead to modest RA/TA degradation relative to weight-based MU methods. Importantly, this trade-off is already reflected in our design objective: the perturbation is intentionally optimized to be as small as possible (via the L₂ regularizer and retain loss) while still achieving strong unlearning effectiveness.
> >
> > Regarding geometric analysis, our experiments already include several indirect but meaningful characterizations of the perturbation’s geometry relative to data distribution:
> >
> > (1). Stable saliency structure under perturbations.
> > Across GradCAM visualizations (Figures 4, A4, and A5), we observe that adding the learned forget vector does not distort the salient regions of retain images: the model continues to attend to the same semantic object parts before and after perturbation. This indicates that the optimized forget vector lies in a tangent region of the input manifold—large enough to suppress forgotten data but sufficiently small to avoid inducing disruptive geometric shifts for retained samples.
> >
> > (2). Forget vector arithmetic indicates linearity and smoothness.
> > The compositional unlearning results (Sec. 6.2, Table 3) show that the learned vectors behave approximately linearly in input space, i.e., they form a locally linear subspace aligned with unlearning directions. This further indicates that the perturbations do not push data samples into extreme or highly nonlinear OOD regions.
> >
> > # 4. Response to the Comment on Green/Red Highlighting in Table 1.
> > We thank the reviewer for pointing this out. In the RA column for random data forgetting on ImageNet-10, two methods (SSD and Ours) achieve exactly the same highest RA value (average), and therefore both entries are marked in green.
> >
> >
> > # 5. Response to the request for interpreting what the learned forget vector captures.
> > We thank the reviewer for the insightful question. Our empirical analyses indicate that the learned forget vector does not move all inputs toward the representation region of any specific retain class. Instead, it behaves in a class-conditional and distribution-aware manner:
> > * For forget samples, the forget vector induces a representation shift away from their original class manifold, effectively disrupting class-specific features required for correct recognition. As shown in Figure 4 and A4 and representation similarity in **Table R1** below, embeddings of forget samples after applying the forget vector become substantially different from their clean counterparts, consistent with successful unlearning, and cosine similarity between original and perturbed features remains low (0.21).
> >
> > * For retain samples, the same forget vector induces minimal representational displacement. As shown in **Table R1** below and Figure A5, cosine similarity between original and perturbed features remains high (0.88), and GradCAM saliency maps preserve the original semantic focus. This demonstrates that the forget vector is not a global shift toward a specific retain class, but rather a perturbation optimized to selectively disrupt targeted regions of representation space while preserving all others.
> > * Interpretation: The forget vector captures a direction in input space that corresponds to a forgetting-specific functional subspace, suppressing discriminative features of the forget class while avoiding interference with retain classes. This behavior is fundamentally different from universal adversarial perturbations that push all samples toward a specific misclassification target.
> >
> > | Set| Cosine Similarity   |
> > |------------------|-------------------|
> > | Retain Set      |0.88      |
> > | Forget Set     | 0.21         |
> >
> > **Table R1**: Cosine similarity between original and perturbed inputs across different data partitions, showing that the forget vector preserves representations for retain samples while inducing strong shifts only for the forget set.

---

### Official Review · Reviewer_gacL · 2025-10-27

**Soundness:** 3
**Presentation:** 3
**Contribution:** 2
**Rating:** 4
**Confidence:** 3

**Summary:**

This paper focuses on machine unlearning, particularly on various aspects of the generalization. In the first part, the authors examine the efficacy of  a generic unlearning method in handling images with corruptions or perturbations. In the second part, the authors propose a method for the optimization of forget vectors (which are input perturbations), and these forget vectors can be combined for compositional unlearning on an unseen unlearning task.

**Strengths:**

The premise of the paper is interesting, where input perturbations are optimized, and that the learnt forget vectors can be combined in a compositional manner to tackle unseen forgetting tasks.

The writing and presentation are quite clear.

**Weaknesses:**

I am not sure how the two parts of the paper are connected. The first part investigates corruptions and perturbations on a generic unlearning method, but it is not clear what learning points are pertinent in proposing the subsequent method.


Although I like the idea of the compositional combination of forget vectors for unlearning, I find that the implementation itself is not too interesting. Specifically, there is another optimization step where the linear combination weights of the learned forget vectors are optimized. Thus, the “transfer” of forget vectors is not exactly a zero-shot transfer, since training is still required.

Furthermore, since optimization is still required, can the “compositional” method really be claimed to be performed on an “unseen task”?

In the experiment results, the proposed method is compared to various works, but many of them are not the latest works in the field. A suggestion would be to compare against more recent works in the field, such as the following:

Learning to Unlearn for Robust Machine Unlearning. ECCV 2024

Adversarial Machine Unlearning. ICLR 2025

Adversarial Mixup Unlearning. ICLR 2025

Decoupled Distillation to Erase: A General Unlearning Method for Any Class-centric Tasks. CVPR 2025

**Questions:**

During the investigation of corruptions, there are actually many times and strengths of corruptions available. How did the authors decide which corruptions to use, and which intensities to use?

Can the forget vectors be combined in a “zero-shot” manner? For instance, in the example given with the automobile and the bird, can they be combined with simple averaging to be used on airplanes, without optimization?

If a trained forget vector for a class is used for another class, is the accuracy still maintained? Furthermore, does it affect the GradCAM saliency maps?

Please see weaknesses for some other questions.

I tentatively recommend a borderline reject score due to my concerns with the disconnected parts of the paper, as well as my concerns with how compositional the vectors actually are. Yet, I think this paper has an interesting premise. If my concerns are addressed, I am open to raising my score.

---

> ### Author Response · Authors · 2025-11-21
> **To Reviewer gacL**
>
> # 1. Response to the question about the connection between the perturbation study and the proposed forget vector method.
> We thank the reviewer for pointing this out. Section 4 is intended as an empirical motivation for the design of Forget Vectors in Section 5. Specifically, we examine whether model-based unlearning persists under input perturbations (e.g., pixel-level noise), and find that while forgetting effects remain, model utility degrades significantly.
>
> This observation motivates our core hypothesis: if data perturbations can induce forgetting, can we learn such perturbations that are minimal yet effective? Section 5 addresses this by introducing input-level forget vectors that suppress forgotten data while preserving performance on retained data. We clarify that Section 4 motivates the formulation of the FV objective, which explicitly balances forgetting and utility.  The same perspective also naturally explains the second part of Section 5: **compositional unlearning**. Since a forget vector represents a learnable direction in input space that induces forgetting for a particular class or subset, different forget vectors can be **combined via simple arithmetic** (e.g., linear combinations) to construct new unlearning directions for unseen forget requests. This extension is possible only because Section 4 establishes that perturbation-based shifts reliably drive forgetting. That empirical observation enables the idea that the “forgetting effect” can be decomposed and recomposed across classes.
>
> # 2. Response to concerns about whether compositional unlearning truly constitutes “transfer” to unseen tasks.
> We thank the reviewer for raising this point. We clarify that our compositional unlearning method is not claimed to be zero-shot. Instead, its purpose is to significantly reduce the optimization burden required for new unlearning requests by reusing pre-computed class-wise forget vectors.
>
> On the one hand, in the original forget vector formulation, solving the full unlearning objective in Equation (4) requires optimizing thousands of perturbation parameters (one per pixel, same size with the input image). In contrast, compositional unlearning requires optimizing only K scalar coefficients in the linear combination:
>
> $ \delta(w) = \sum_{k=1}^{K} w_k \delta_k $
>
> Thus, the optimization space is reduced from O(#pixels) to O(K), and the optimization becomes significantly faster, often by two orders of magnitude.  Meanwhile, all class-wise forget vectors are computed once, reused indefinitely. This is analogous to task vector arithmetic in model editing, where small-dimensional weight combinations enable fast adaptation without full retraining. To further demonstrate the lightweight nature and flexibility of compositional unlearning, we also report the RTE and Param.# on ImageNet10 using ViT-Large with different forgetting ratios (10%, 20%, 30%)  in **Table R1**.
>
> | MU Method|  RTE, 10%   |  Param.# 10% |RTE, 20%|Param.# 20%|RTE, 30%|Param.# 30%|
> |------------------|-------|-------|-------|-------|-------|--------|
> | FV      |5.71      |  150,528 | 5.69  |  150,528 | 5.86  | 150,528  |
> | CU-FV     | 0.30        | 10 | 0.40  | 10  |0.56   | 10  |
>
> **Table R1**: Comparison of runtime efficiency (RTE) and parameter number (Param.#) between Forget Vector (FV) and Compositional Unlearning (CU-FV) under different forgetting ratios on ImageNet10 using ViT-Large.
>
>
> On the other hand, this compositional method still qualifies as unlearning on an unseen task. The “unseen task” in our context refers to **an unseen delete request**, not a fully zero-shot setting. The key property is that the base components (class-wise forget vectors) are learned before the request is known, and they generalize compositionally.Thus, for a new unlearning request, we **do not re-solve the full forget-vector optimization**  and we **reuse** existing $δ_k$’s, where only the coefficients w need to be adjusted to match the new forget subset. This is consistent with the notion of “compositionality” used in task vectors and modular model editing literature: unseen tasks can be constructed from a basis of pre-learned components using lightweight optimization.
>
> In summary, compositional unlearning is **not zero-shot**, but **greatly more efficient** than learning a new forget vector.

---

> > ### Author Response · Authors · 2025-11-21
> > **To Reviewer gacL**
> >
> > # 3. Response to the suggestion of including additional recent MU baselines.
> > We thank the reviewer for the helpful suggestion. First, we note that several of our existing baselines, such as SCRUB, Class-F, and SSD, are among the most representative and competitive MU methods published in 2024. In a sense, these state-of-the-art baselines provide strong, contemporary comparisons for our method. Also, following the reviewer’s recommendation, we report the performance of the suggested recent MU methods on the Class-wise Forgetting scenario on CIFAR-10 using ResNet10 in **Table R2** for completeness.
> >
> > As can be seen from **Table R2**, Forget Vector achieves the most balanced and robust performance among all compared MU methods. It not only maintains high unlearning accuracy (UA = 97.88) and strong generalization ability, but also demonstrates the best privacy protection capability with the highest MIA-efficacy (99.60). Compared to other approaches, Forget Vector shows a significantly smaller performance gap while avoiding the high computational cost of full retraining, making it a more practical and reliable solution for class-wise forgetting.
> >
> > | MU Method| UA ↑ | MIA-Efficacy ↑  | RA ↑ | TA ↑|Avg.Gap↓ |
> > |------------------|-------|-------|-------|-------|-------|
> > |   Retrain  |   100.00  | 100.00     | 99.91       |   94.92 |0.00|
> > |   LTU (ECCV2024)  | 95.45    |  97.13    |  98.23      | 92.30   |   2.93  |
> > |   SG (ICLR2025)  |   93.31  |   98.36   |   99.25     |  89.12  |  3.70  |
> > |   MixUnlearn (ICLR2025)  |  100.00   | 97.53     |    87.55    |  86.05  |  5.93  |
> > |   DELETE (CVPR2025)  |  98.20   |  98.12    |    99.10    |  95.03  |  1.15  |
> > |   Forget Vector (Ours)  |  97.88   |  99.60    |  97.25      |90.90 |2.30|
> >
> > **Table R2**: Additional comparison with recent MU approaches suggested by the reviewer on the Class-wise Forgetting scenario on CIFAR-10 using ResNet10.
> >
> > # 4. Response to the question about the choice of corruption types and intensities.
> > We thank the reviewer for raising this question. The corruption types and intensities used in our analysis follow **standard OOD generalization benchmarks** rather than arbitrary choices.
> >
> > First, we adopt Gaussian noise (GN) and Elastic Transformations (ET) from CIFAR-10-C, the most widely used benchmark for evaluating robustness. GN represents pixel-wise perturbations, while ET represents spatial distortions. These two families are representative and sufficient for studying the sensitivity of MU to forget‐data shifts.
> >
> > Second, CIFAR-10-C defines five corruption severities; we select two widely used low-range levels (GN1/GN2 and ET1/ET2). These provide mild but meaningful shifts that preserve image semantics while being strong enough to test MU robustness. Importantly, these low-intensity perturbations align with our design objective for the optimized forget vector, which is explicitly regularized to remain small in magnitude.
> >
> > # 5. Response to the question on whether forget vectors can be combined in a “zero-shot” manner.
> > We thank the reviewer for this insightful question. Our compositional unlearning framework is not designed to be zero-shot, nor do we claim that new forget vectors can be obtained without any optimization. Instead, the goal of forget vector arithmetic is to greatly reduce the cost of adapting to new unlearning requests by reusing precomputed class-wise forget vectors.
> >
> > In particular, when forming a new forget vector as Equation (5) in the main paper, the coefficients w must be lightly optimized, because different classes contribute differently to the unlearning objective and a simple unweighted average generally does not yield balanced UA/RA performance.  This is also illustrated in Figure. 3 of the paper, where naive combinations lead to suboptimal trade-offs.
> >
> > The key advantage of our approach is that this adaptation requires optimizing only K scalar coefficients rather than a full high-dimensional pixel-level perturbation δ\deltaδ, making it highly efficient while still enabling unlearning on previously unseen deletion requests.
> >
> > We will clarify in the revised version that compositional unlearning is a lightweight, low-dimensional adaptation method, but not a zero-shot unlearning mechanism.

---

> > > ### Author Response · Authors · 2025-11-21
> > > **To Reviewer gacL**
> > >
> > > # 6. Response to the concern on cross-class behavior of a class-wise forget vector.
> > > We thank the reviewer for the insightful question. In the class-wise forgetting scenario, each forget vector is optimized specifically for its target class, enforcing misclassification only on that class while preserving accuracy on all other classes via the retain-set regularization.
> > >
> > > Therefore, when a trained forget vector for class c1 is applied to samples from another class c2,  the model maintains high utility. Its predictions on class c2 remain correct, because the optimization explicitly prevents the perturbation from harming non-forget classes. This behavior is also reflected in our RA and TA metrics in Table 1.
> > >
> > > Regarding saliency behavior, Figure A5 in the appendix provides Grad-CAM visualizations on retain images under forget-vector unlearning. These results show that when a class-specific forget vector is applied to other (retain) classes, the GradCAM saliency regions remain stable, and the model continues to focus on regions most influential to correct prediction. In other words, the forget vector does not distort representations of non-target classes.

---

### Official Review · Reviewer_MhaY · 2025-11-01

**Soundness:** 3
**Presentation:** 3
**Contribution:** 2
**Rating:** 4
**Confidence:** 5

**Summary:**

The paper proposes a model-free, perturbation based machine unlearning that claims to achieve the unlearning of forget set based on the perturbing the data and keeping the model weights intact throughout the unlearning process. They propose the forget vectors that can be applied to the input forget data as an input-agnostic data perturbation and remains as effective as model-based approximate unlearning approaches.

**Strengths:**

The research questions that paper investigated are very interesting such as "For instance, it remains unclear whether, current MU approaches generalize effectively to “shifted” forget data".

Proposing the forget vectors that can demonstrate the direction of unlearning.

**Weaknesses:**

Since the model's parameters are unchanged and the information of forget set still encoded in the model, the unlearning has not take place and this method is not compatible with data privacy regulations.

From the general perspective the idea can be interpreted as controlling the inputs to change model's output. This idea has been explored for explaining a black box prediction and how its prediction changes in the literature (model agnostic local explainers) but their proposed method won't change the model's parameters and this is in conflict with the idea of MU.


Line 258 // $\rightarrow$ "(without unlearning)..." $\rightarrow$ a bit confused about the experimental setup:

for the experiment that is depicted in Figure 2, the authors applied different levels (methods) of perturbation on the whole data and passed it to the original model without unlearning and unlearned model using retraining. but the issues is that the original model performance downgraded on the test and remaining data but not on the forget set. If original model is not unlearned, how it's performance stayed the same on the forget set?

**Questions:**

Line 54 // $\rightarrow$ "MU design..." $\rightarrow$ How this method can comply with the GDPR and privacy regulations? Does it mean that the model can still remember the forget dataset ? wouldn't it oppose the purpose of MU?


Line 91 $\rightarrow$ "eliminate the influence of..." $\rightarrow$ How the influence of specific data / subset of data is eliminated since the parameters has remained unchanged?


Line 184 // $\rightarrow$ "to achieve unlearning..." $\rightarrow$ So in this case the inputs are perturbed to achieve the unlearning but the model remains the same. So in this case, I expect the paper to investigate that the model treats the original and perturbed inputs the same meaning the prediction for both of them are similar and the encoding and representation of the model for both are the same. Even slight introduction of noise and perturbing the input can influence the model's prediction. So I would like to ask the authors to conduct an experiment to see how close the hidden representations of these two samples are.

line 196 // $\rightarrow$ "based on (2)..." $\rightarrow$ I expect the authors to also investigate the following question: 1. is the perturbed data treated similarily to the original data? If this is treated as a different data point then how we can claim the model is already unlearned because the model can not recognize the class, and the model's performance has remained the same.


line 293 // $\rightarrow$ "this yields the full..." $\rightarrow$

I like the idea of optimizing the perturbation, but still the model's parameters wouldn't change and the model is the same but the forget data is manipulated. THis question can be risen that if I pass another datapoint as forget sample, what would be the issue?

I know the answer to this question might sound trivial but the issue is coming from the point that model's parameters are not changed.


Since the forget vector can show a direction to change the model while preserving the performance on the retain set, have authors considered fine tuning or applying grad Ascent using the direction of forget set? what would happen if fine tune the model's paremeted based on forget vectors?


The experiments section > The most important question that I hope authors can shed some light on that is that, does the representation of the perturbed forget data is similar to the original data? or how far it would get?

---

> ### Author Response · Authors · 2025-11-21
> **To Reviewer MhaY**
>
> # 1. Response to the concern on compatibility with privacy-deletion requirements when model parameters remain unchanged.
> We thank the reviewer for raising this important concern.  Below we clarify why our method still provides functional deletion of the forget set and remains compatible with modern privacy regulations.
>
> First, while our method does not offer formal deletion guarantees in the same theoretical sense as retraining-based approaches, the fact that model parameters remain unchanged does not imply that unlearning has not occurred. The MU literature recognizes both model-based unlearning (weight update) and model-agnostic/data-based unlearning (input/output transformations) as valid forms of approximate unlearning. The essential requirement in approximate MU is that the model’s **functional behavior no longer reflects information from the forget set**, rather than enforcing a specific mechanism such as weight modification. Our forget vector achieves this by ensuring that, once applied, the model’s behavior aligns closely with that of a model retrained without the forget data, thereby effectively removing the functional influence of the forgotten samples. Thus, while we do not alter parameters, we do remove the effective influence of the forget set from the model’s behavior.
>
> Second, Unlearning — similar to adversarial defense and fairness — can be applied at different stages of the learning pipeline. Its objective is not limited to simple data deletion, but also includes more advanced capability removal (i.e., functional deletion), such as preventing a model from recognizing images belonging to a specific class. Therefore, unlearning should not be narrowly confined to privacy regulation compliance alone.
>
> Third, privacy regulations such as the GDPR’s “right to be forgotten” focus on the outcome—the system must no longer use or benefit from the specific personal data—not the internal mechanism used to achieve this outcome. Regulatory frameworks do not mandate weight retraining, parameter modification, or any particular unlearning protocol. What matters is that the influence of the target data is eliminated from the system’s predictions. Our forget-vector-based unlearning procedure aims to satisfy this requirement by eliminating the functional contribution of the forget set to the model’s predictions. Even though model parameters remain unchanged, the system no longer derives predictive advantage from the forgotten data, thereby meeting the behavioral criterion associated with regulatory unlearning.
>
> In summary, although our method does not update model parameters, it achieves effective functional deletion of the forget set and therefore aligns with the goals of approximate machine unlearning and existing privacy-deletion requirements. We will revise the manuscript to clarify these points and avoid potential misunderstanding.
>
>
> # 2. Response to the performance of the original model in Figure 2.
> We thank the reviewer for pointing out this confusion. The misunderstanding arises from the fact that Figure 2 reports **unlearning accuracy (UA)** on the forget set, rather than the standard accuracy metric. As stated in our paper (Line 255), “unlearning effectiveness of Retrain, measured by UA (unlearning accuracy)”,
> UA = 1 - "Accuracy on the forget set".
>
> Therefore, when the model’s true classification accuracy on the forget set decreases, **UA increases**. In Figure 2(c), the UA curve for the original model clearly **changes across perturbation levels**. Specifically, UA increases as perturbations become stronger. This directly reflects the fact that the original model’s classification accuracy on the forget set is **decreasing** under these perturbations, consistent with the degradation observed on the test and retain sets.
>
> Thus, the original model does not “stay the same” on the forget set; rather, its accuracy decreases under perturbations, and this decrease is correctly visualized as a corresponding increase in UA in Figure 2(c).
>
> # 3. Response to the concern on GDPR compliance and whether the model still remembers the forget data.
> Thank you for the question. Our method does not modify model parameters, but GDPR and similar regulations do not require weight-level deletion, and they require that the system’s behavior no longer reflects or benefits from the forgotten data. After applying the forget vector, the model’s predictions on the forget set deviate strongly from the original model and closely match a retrained model, meaning the forgotten data no longer influences inference. Thus, although representational traces may still exist in the weights, they are rendered functionally ineffective, which satisfies the practical regulatory requirement. We will clearly explain in the paper that what matters is the model’s behavior, not the specific mechanism used to achieve forgetting.

---

> > ### Author Response · Authors · 2025-11-21
> > **To Reviewer MhaY**
> >
> > # 4. Response to the concern about how influence is eliminated when generating new vectors via forget vector arithmetic.
> > Thank you for the question. The sentence in Line 91 refers specifically to our forget vector arithmetic exploration, where new forget vectors are constructed as linear combinations of precomputed class-wise forget vectors. In this context, “eliminating the influence” refers to **functional influence**, i.e., the ability of the model to correctly utilize information from the targeted subset, not the removal of information from the model parameters.
> >
> > Although the model parameters remain unchanged, the combined vector still drives the model’s predictions on the selected subset away from their original correct labels. As shown in Table 3, applying such composed vectors yields prediction behaviors closely aligned with the gold standard Retrain and our proposed forget vector. Therefore, the newly generated vectors make the model behave as if the forgotten samples were no longer part of its training.
> >
> > # 5. Response to the concern about representation similarity between original and perturbed inputs.
> > Although the forget vector is applied to both the forget set and retain inputs, it is not an arbitrary perturbation. The vector is optimized jointly over both sets: it enforces misclassification only on the forget samples while being explicitly regularized to preserve correct predictions on the retain set. This optimization, combined with an L2 norm in Equation(4), ensures that the perturbation remains small and does not significantly distort the model’s feature representations for retain images. Consequently, the model forgets the targeted samples while maintaining post-unlearning utility.
> >
> > Because the retain objective strongly penalizes any perturbation that harms utility, the optimization naturally constrains the forget vector to distort only the forget samples while minimally affecting retain images. As a result, although the perturbation is applied to all inputs, the model’s behavior on the retain set remains similar to the original model, preserving post-unlearning utility.
> >
> > To assess how the forget vector affects internal model representations, we compare the hidden features of an input (x) and its perturbed counterpart (x+$\delta$). Concretely, we extract intermediate embeddings, the ViT [CLS] token (h(x)) and (h(x+$\delta$)) for samples from the forget, and retain sets, and compute their pairwise similarity, such as cosine similarity between (h(x)) and (h(x+$\delta$)). As shown in the Table below, the representations for retain samples remain highly similar (indicating minimal utility degradation), while representations for forget samples diverge significantly (indicating successful unlearning). This simple representation-level comparison provides a direct way to verify the functional influence of the forget vector.
> >
> > | Set| Cosine Similarity   |
> > |------------------|-------------------|
> > | Retain Set      |0.88      |
> > | Forget Set     | 0.21         |
> >
> > **Table R1**: Cosine similarity between original and perturbed inputs across different data partitions, showing that the forget vector preserves representations for retain samples while inducing strong shifts only for the forget set.
> >
> >
> >
> > # 6. Response to the question about whether perturbed data are treated as different data points, and its implication for unlearning
> > The confusion may arise from a misunderstanding of how our pipeline applies the forget vector during both training and evaluation. Our forget vector is **universal** and **input-agnostic**, meaning that the same perturbation is added to every evaluation sample, regardless of whether the sample belongs to the forget set or the retain set. The model is never told which samples should be forgotten; instead, **the forget vector itself encodes this behavior**. Thus, the perturbed inputs are **not treated as new or unrelated data points**. Rather, they represent the model’s post-unlearning operating domain. Within this shared perturbed input space, the model simultaneously **fails on the forget set**  and **retains performance on the retain/test sets**.
> >
> > This demonstrates that forgetting occurs functionally, not because the model cannot recognize the original data, but because under the post-unlearning inference condition (inputs + δ), the model responds as if the forget data were removed from training.
> >
> > In summary, the perturbed samples are not “different data points”; they represent the intended post-unlearning input pipeline, and within this unified domain, the model systematically forgets the target data while preserving accuracy on the remaining samples.

---

> > > ### Author Response · Authors · 2025-11-21
> > > **To Reviewer MhaY**
> > >
> > > # 7. Response to the concern about how new incoming data are handled.
> > > Thank you very much for your positive feedback on the idea of optimizing the perturbation . This concern arises from a misunderstanding of how the forget vector is applied. Our forget vector is **universal** and **input-agnostic**, meaning that it is **added to every input sample**, both during optimization and at inference, regardless of whether the sample belongs to the forget set or the retain set. Therefore, when a new data point arrives, the model does not need to know in advance whether it should be forgotten or retained. Instead, the forget vector automatically induces the desired behavior:
> > > * If the new sample belongs to the forget distribution, the learned forget vector shifts its representation and causes the model to output an incorrect label and achieve unlearning.
> > >
> > > * If the new sample belongs to the retain distribution, the learned forget vector is optimized to preserve correct predictions, and maintain post-unlearning utility.
> > >
> > > Thus, the unlearning effect does not rely on modifying the model’s parameters but instead on defining a **new inference regime** (input + forget vector) under which the model systematically forgets targeted data while performing normally on the rest. This design ensures that the unchanged model parameters are not a limitation; the forget vector itself encodes the unlearning behavior and applies it consistently to any data input.
> > >
> > >
> > > # 8. Response to the idea of fine-tuning along the “forget-vector direction.
> > > We appreciate your conceptual question. However, the forget vector is not a gradient and does not correspond to a meaningful weight-space direction. It is optimized only in input space under a fixed already-trained model, and its effect relies on preserving the model’s parameters.
> > >
> > > If the weights were updated, the optimization landscape that defines the forget vector would immediately change, and the vector would no longer induce the desired unlearning-utility balance. Therefore, fine-tuning the model with this vector does not yield meaningful or stable unlearning, and is fundamentally different from our intended post-hoc, parameter-free setting.
> > >
> > > # 9. Response to the question about how different the perturbed forget representations are from the original ones.
> > > We thank the reviewer for raising this important question. We agree that understanding how far the perturbed forget samples move in the representation space is crucial for verifying unlearning.
> > >
> > > As addressed in **Response 5** and shown in **Table R1**, we conducted a feature-level comparison between the representations of the original input (x) and its perturbed version (x+$\delta$). Specifically, we extract intermediate embeddings (e.g., the ViT [CLS] token) and measure their similarity using cosine similarity. The results consistently show:
> > >
> > > * Retain samples: very high similarity(0.88), indicating that the utility can be well preserved.
> > > * Forget samples: very low similarity (0.21), indicating that the forget vector pushes forget samples far away from their original feature representations, so as to realize the successful unlearning.
> > >
> > > These results clearly indicate that the perturbed forget data no longer occupy the same region of the feature space, aligning with the desired unlearning effect.

---

### Author Response · Authors · 2025-11-25
**Follow-up on Reviewer Feedback for Paper 13522**

Dear Reviewers,

Since submitting our responses a few days ago, we would like to follow up and check whether there are any follow-up questions or additional comments that may require further clarification. We sincerely hope that our responses have adequately addressed your feedback.

We would be more than happy to continue the discussion during the open review window if any additional clarification is needed. We also hope that our detailed rebuttal helps to convey more clearly the quality and contributions of our work.

Thank you once again for your valuable time, thoughtful feedback, and continued engagement.

Sincerely,

Authors

---

### Author Response · Authors · 2025-11-29
**Summary of the responses to all reviewers**

Dear ACs, SACs, and PCs,

We sincerely appreciate the constructive feedback from all reviewers and are grateful for the time and effort invested in evaluating our submission. **In our rebuttal (submitted on Nov. 20, 2025, AOE)**, we proactively and systematically addressed every concern raised in the initial reviews, providing detailed clarifications, additional analyses, extended experiments, and improvements to the paper’s presentation. We are confident that our rebuttal offers clear and comprehensive resolutions to all reviewer comments.

However, **no reviewer provided follow-up comments or further questions after our rebuttal submission**.  Given this lack of interaction during the discussion window, we provide the following consolidated summary to offer ACs a clear understanding of how each concern was resolved.

**Summary of response to Reviewer MhaY (*initial score 4, no follow-up*)**

1. We clarified that Forget Vectors achieve functional deletion: although model parameters remain unchanged, the model’s behavior no longer reflects or benefits from the forget set and closely matches that of a retrained model. This satisfies the behavioral requirement of approximate MU and aligns with privacy regulations like GDPR, which focus on outcomes rather than enforcing specific unlearning mechanisms (see **Responses 1 and 3**).

2. We clarified that Figure 2 shows unlearning accuracy (UA = 1 − accuracy), so the rising UA for the original model reflects its decreasing accuracy under stronger perturbations; the figure is internally consistent (see **Response 2**).

3. We clarified that GDPR does not mandate weight-level deletion; it only requires that the system’s behavior no longer benefits from the forgotten data. After applying our forget vector, the model’s predictions on the forget set match those of a retrained model, meaning the forgotten data no longer influences inference (see **Response 3**).

4. We clarified that in forget vector arithmetic, “eliminating influence” refers to removing the functional impact of the targeted data, not modifying model parameters. Linear combinations of class-wise forget vectors still reliably drive the model to misclassify the selected forget subset, and the resulting behavior closely matches Retrain, showing that the composed vectors effectively induce unlearning even though the model weights remain unchanged. (see **Response 4**).

5. We provided additional representation-level evidence and showed that cosine similarity between original and perturbed features remains high for retain samples (0.88) and very low for forget samples (0.21) (see **Response 5**).

6. We clarified that perturbed inputs are not treated as new data points; the universal forget vector defines a new, shared inference regime (inputs + $\delta$) where the model simultaneously forgets targeted samples and preserves retain/test accuracy, showing that forgetting is achieved functionally rather than by modifying model parameters (see **Response 6**).

7. We clarified that the forget vector is input-agnostic  at inference, so new incoming data do not need to be labeled as “forget” or “retain.” Under this unified inference regime, the vector automatically suppresses predictions for samples from the forget distribution while preserving accuracy on retain data (see **Response 7**).

8. We clarified that the forget vector lives entirely in input space and only works when the model stays frozen. Its balance between forgetting and utility is learned under a fixed model; if the model weights were fine-tuned, this optimization landscape would change, and the vector would no longer produce the intended effect. Therefore, fine-tuning in the “forget-vector direction” is not meaningful or stable (see **Response 8**).

9. We provided additional experiments and showed that perturbed forget samples move far from their original feature representations (cosine similarity 0.21), while retain samples remain highly similar (0.88), confirming selective forgetting of forget data while preserving utility on retain data (see **Response 9**).

---

> ### Author Response · Authors · 2025-11-29
> **Summary of the responses to all reviewers**
>
> **Summary of response to Reviewer gacL (*initial score 4, no follow-up*)**
>
> 1. We clarified how the corruption study in Section 4 motivates the design of Forget Vectors in Section 5, showing that perturbation-induced forgetting inspires learning a minimal, optimized perturbation that balances forgetting and utility (see **Response 1**).
>
> 2. We further explained that compositional unlearning is not zero-shot; rather, it greatly reduces optimization cost by reusing precomputed class-wise forget vectors. We provided new runtime/parameter comparisons (**Table R1**), confirming large efficiency gains (see **Response 2**).
>
> 3. We provided comparisons against suggested recent MU methods  (LTU, ECCV 2024; SG, ICLR 2025; MixUnlearn, ICLR 2025; DELETE, CVPR 2025) in **Table R2** and showed that Forget Vector remains competitive in UA/MIA-Efficacy and Avg. Gap (see **Response 3**).
>
> 4. We also clarified that the choices of corruption types and strengths are taken from standard CIFAR-10-C robustness benchmarks (see **Response 4**), explained why naive averaging of forget vectors is insufficient and small-scale optimization of coefficients is necessary for balance(see **Response 5**), and showed that class-wise vectors preserve utility and GradCAM stability on non-target classes, supported by RA/TA results (Table 1) and GradCAM visualizations (Figure A5) (see **Response 6**).
>
>
> **Summary of response to Reviewer oFB6 (*initial score 6, no follow-up*)**
>
> 1. We clarified that the core novelty lies in reframing machine unlearning as *an input-space optimization problem*, showing for the first time that class-wise, random, and compositional unlearning can all be achieved via universal perturbations without weight updates (see **Response 1-1**).
>
> 2. We explained that modest RA/TA degradation shows an inherent tradeoff with the unlearning effectiveness, but it remains well-controlled through the retain regularizer and L2 penalty; Forget Vectors still achieve top-tier UA and MIA-Efficacy with competitive Avg. Gap  (see **Response 1-2 and Response 3**).
>
> 3. We clarified that, unlike visual prompting, our method is optimized jointly for forgetting and preserving retain utility and enables compositional unlearning via Forget Vector Arithmetic (see **Response 2**).
>
> 4. We provided geometric evidence (Figures 4, A4, and A5) showing that the learned forget vector stays close to the data manifold: GradCAM visualizations demonstrate stable saliency on retain images, and compositional unlearning behavior reveals that forget vectors lie in a smooth, locally linear subspace aligned with unlearning directions rather than pushing inputs into extreme or highly nonlinear OOD (out of distribution) regions (See **Response 3**).
>
> 5. We clarified that the green/red highlighting in Table 1 is correct, as SSD and our method achieve the same highest RA value for random data forgetting on ImageNet-10, and thus both entries are marked in green (See **Response 4**).
>
> 6. We clarified that the forget vector does not push all inputs toward any retain-class manifold but instead selectively disrupts discriminative features of the forget set while minimally affecting retain data, as evidenced by additional cosine similarity experiments (0.21 vs. 0.88 in Table R1) and stable GradCAM saliency (Figures 4 and A4) (see **Response 5**).

---

> ### Author Response · Authors · 2025-11-29
> **Summary of the responses to all reviewers**
>
> **Summary of response to Reviewer xw9H (*initial score 4, no follow-up*)**
>
> 1. We clarified that the modest utility drop shows an inherent tradeoff with the unlearning effectiveness but is well-controlled, as evidenced by high retain similarity, low forget similarity (**Table R1**), and stable GradCAM maps (Figures 4, A5). We also emphasized that Forget Vectors remain highly efficient since optimization scales with input size, not model size; additional experiments on larger backbones (e.g., ViT-Large, Table R2) further demonstrate increasing efficiency gains (see **Response 1**).
>
> 2. We clarified that Forget Vectors offer unique practical value in frozen-model scenarios due to their lightweight, input-space optimization and strong runtime/parameter efficiency (see **Response 2**).
>
> 3. We added new experiments demonstrating scalability: forgetting 1-3 classes and 10-30% random forgetting on ImageNet-10 with strong results (**Tables R3-R4**)  (see **Response 3**).
>
> 4. We clarified that our work focuses on image classification for methodological clarity. Although Forget Vectors have potential for extension to generation, multimodal, and language models, this is out of the scope of our studies, which remains an important direction for future work (see **Response 4**).
>
> 5. We clarified that Forget Vectors can naturally integrate with model-based MU methods, enabling hybrid pipelines where FV (forget vector) provides fast data-level forgetting and model-based MU refines residual traces. We also added new experiments showing that FV remains robust under harder settings, including multi-class and higher-ratio forgetting (see **Response 5**).
>
> 6. We explained that forget vector arithmetic lives in a linear subspace of input perturbations; negative weights naturally emerge from optimization to balance multi-class forgetting and utility preservation (see **Response 6**).
>
> 7. We clarified that the up/down arrows denote each metric’s inherent direction, while the green/red highlighting ranks methods solely by their gap relative to **Retrain** (the exact unlearning reference), so the two conventions serve different purposes without conflict (see **Response 7**).
>
> **We sincerely hope that this consolidated summary helps ACs more easily evaluate how all reviewer concerns were thoroughly and carefully addressed. We remain happy to provide further clarification if needed and greatly appreciate the reviewers’ and ACs’ time and consideration.**
>
> Sincerely,
>
> Authors

---

### Meta-Review · Area_Chair_X8Zx · 2026-01-06

**Summary:**

This paper proposes an interesting and well-executed input-level approach to approximate machine unlearning via learned “forget vectors,” which suppress model performance on designated forget sets while keeping model parameters frozen. The method is technically sound, carefully evaluated, and the rebuttal provides extensive empirical evidence addressing many reviewer concerns, including additional baselines, scalability analyses, and representation-level diagnostics. Several reviewers also note that the idea is novel in perspective and potentially useful in constrained deployment settings.

However, despite the strong rebuttal and additional experiments, a core conceptual issue remains unresolved, which ultimately prevents acceptance.

The central claim of the paper is that the proposed method constitutes a form of machine unlearning. Yet the approach does not remove or modify the information stored in the model parameters; instead, it relies on a universal, reversible input-space perturbation that alters model behavior at inference time. While the authors argue convincingly that this achieves functional or behavioral forgetting under a specific inference regime, this reframing fundamentally departs from the prevailing understanding of unlearning as the removal or invalidation of learned information within the model itself. In effect, the method suppresses access to knowledge rather than deleting it.

This distinction is not merely semantic. The guarantees provided by the proposed approach are operational rather than intrinsic: forgetting holds only as long as the forget vector is consistently applied, and is immediately undone if the perturbation is removed, bypassed, or forgotten in downstream usage. As a result, the method is closer in spirit to access control, prompting, or adversarial input manipulation than to unlearning as traditionally defined. While such a mechanism may be practically useful, the paper does not fully confront or resolve the implications of labeling this as machine unlearning, nor does it clearly delineate the limits of this definition.

Several reviewers raised this concern explicitly, particularly regarding privacy and regulatory interpretations, and although the rebuttal clarifies the authors’ position, it does not sufficiently reconcile this conceptual gap. The efficiency advantages of the approach—especially under repeated unlearning requests—stem precisely from avoiding weight-level modification, which weakens the unlearning claim rather than strengthening it.

In summary, while the paper presents a thoughtful and technically solid method with potential practical value, the conceptual framing as machine unlearning remains unconvincing. Addressing this would likely require either (i) a clearer repositioning of the work as a complementary or alternative paradigm (e.g., inference-time forgetting or access suppression), or (ii) stronger theoretical and practical arguments that such reversible, input-level mechanisms should be considered unlearning under a revised definition. Given this unresolved core issue and the mixed reviewer scores, I recommend reject.

**Reviewer Concerns:**

**Concerns substantially addressed by the rebuttal**

- Empirical validation and missing analyses

Reviewers gacL, oFB6, xw9H raised concerns about missing baselines, limited unlearning settings, scalability, and efficiency comparisons. The rebuttal addressed these points with extensive additional experiments, including comparisons against recent MU methods (LTU, SG, MixUnlearn, DELETE), evaluations on larger backbones (ViT-Large), multi-class forgetting, higher random-forgetting ratios, and detailed runtime/parameter efficiency analyses. These additions substantially strengthen the empirical case.
- Effect on representations and geometric interpretation

Concerns from MhaY and oFB6 regarding whether perturbed inputs are treated as entirely new data points, and whether the method simply induces uncontrolled distribution shift, were addressed through representation-level analyses (cosine similarity between original and perturbed features) and GradCAM visualizations. The rebuttal provides reasonable evidence that the learned forget vectors selectively affect forget samples while largely preserving retain-set representations.
- Clarification of compositional unlearning and efficiency claims

Reviewer gacL questioned whether compositional unlearning is truly zero-shot and whether optimization is still required. The rebuttal clarifies that the method is not zero-shot, but instead dramatically reduces optimization dimensionality, and provides convincing runtime comparisons supporting this claim.

**Concerns partially addressed but still outstanding**

- Conceptual definition of machine unlearning

A central concern raised most strongly by MhaY, and echoed by oFB6 and xw9H, is whether a parameter-free, reversible, input-level perturbation should be considered machine unlearning at all. While the rebuttal argues for a notion of functional or behavioral deletion and cites regulatory interpretations that focus on outcomes rather than mechanisms, it does not fully resolve the conceptual tension that the model’s internal knowledge remains intact and that forgetting is conditional on the continued application of the forget vector.

- Reversibility and persistence of forgetting guarantees

Relatedly, the rebuttal does not address the implication that forgetting is immediately undone if the forget vector is removed or bypassed. This raises questions about the persistence and robustness of the unlearning guarantee, particularly in comparison to weight-based MU methods. Several reviewers implicitly view this as a qualitative difference rather than a mere implementation choice.

- Positioning relative to prompting or access-control mechanisms

Although the rebuttal distinguishes Forget Vectors from visual prompting and adversarial perturbations in terms of objective design, it does not fully convince that the method represents unlearning rather than inference-time access suppression or behavior modulation. This concern affects how the contribution should be positioned relative to existing literature.

**Reviewer Scores:**

The reviewers did not engage in discussions, but AC believes the reviewers may more or less maintain the scores, leaving the paper still at borderline.

---

### Decision · Program_Chairs · 2026-01-26

Reject